# Atypical pericapillary Ly6G+Nur77+ macrophages initiate type-2 immune responses to allergens in the mouse lung

Audrey Meloun [1,2], Holly Bachus[3], Crystal Lewis[4], Brittany Dulek[5], Shivangi Dave[3,6], Dave Durell Hill[7], Gabriela Pessenda[8], Jose Carlos Gonzalez [9], P'ng Loke [8], Alexander F. Rosenberg[7] & Beatriz León [1] ✉

The mechanisms of airway allergen sensing and type 2 immune response initiation remain poorly understood. Using a mouse house dust mite (HDM)-induced allergic airway model, we identify a population of lung macrophages located close to alveolar capillaries that express Ly6G and the nuclear receptor Nr4a1/Nur77. These atypical Ly6G+Nur77+ macrophages preferentially capture airway-delivered allergens and play an important role in initiating HDM-driven T helper type 2 (Th2) responses. They sense the major HDM allergen, the cysteine protease Der p 1, via protease-activated receptor 2 (PAR2), and their activation and accumulation require both PAR2 and Nr4a1/Nur77. These Ly6G+Nur77+ macrophages regulate the migration of conventional migratory dendritic cells (mDCs) to draining mediastinal lymph nodes (mLNs) through cysteinyl leukotriene (CysLT) production, which enhances mDC migration toward CCL21 for T cell priming. Inhibiting CysLT biosynthesis reduces mDC migration and dampens Th2 allergic responses, highlighting possible therapeutic avenues in type 2 immunity.

The mechanisms by which the immune system detects pathogens are well understood, involving pathogen-associated molecular patterns (PAMPs) recognized by pattern recognition receptors (PRRs) that trigger innate immune responses[1]. However, the recognition of common environmental allergens is less clear, as allergens often lack distinct PAMPs. Therefore, understanding how allergens are detected and initiate allergic inflammation remains a critical unanswered question. House dust mites (HDM) are the leading cause of indoor allergic diseases, including asthma and allergic rhinitis, affecting more than 50%

of allergic patients[2]. Despite their clinical significance, the mechanisms by which HDM components are recognized by the immune system and trigger allergic inflammation remain controversial and poorly understood. Lipopolysaccharide (LPS) content in HDM and Toll-like receptor 4 (TLR4) activation have been implicated in immune activation[3,4]. However, other studies have demonstrated that neither LPS nor TLR4 is required and, in fact, suppress rather than promote type 2 allergic inflammation induced by HDM[5–9]. Other research has focused on the protease activity of HDM allergens[2], particularly cysteine protease

[1]Innate Cells and Th2 Immunity Section, Laboratory of Allergic Diseases, National Institute of Allergy and Infectious Diseases, National Institutes of Health, Bethesda, Maryland, USA. [2]Department of Microbiology, University of Alabama at Birmingham, Birmingham, Alabama, USA. [3]Adaptive Immunity and Immunoregulation Section, Laboratory of Allergic Diseases, National Institute of Allergy and Infectious Diseases, National Institutes of Health, Bethesda, Maryland, USA. [4]Department of Pathology-Molecular and Cellular Pathology, University of Alabama at Birmingham, Birmingham, Alabama, USA. [5]Collaborative Bioinformatics Resource, National Institute of Allergy and Infectious Diseases, National Institutes of Health, Bethesda, Maryland, USA. [6]Department of Medicine-Immunology and Rheumatology, University of Alabama at Birmingham, Birmingham, Alabama, USA. [7]Department of Biomedical Informatics and Data Science, University of Alabama at Birmingham, Birmingham, Alabama, USA. [8]Type 2 Immunity Section, Laboratory of Parasitic Diseases, National Institute of Allergy and Infectious Diseases, National Institutes of Health, Bethesda, Maryland, USA. [9]Department of Neurobiology and McKnight Institute, University of Alabama at Birmingham, Birmingham, Alabama, USA. ✉e-mail: beatriz.leonruiz@nih.gov

activity, which has been shown to promote type 2 skin inflammation[10]. In this context, Der p 1, the major allergen in HDM, exhibits potent cysteine protease activity and has been compared to papain, a well-characterized model allergen known to activate type 2 immune responses[11–13]. However, the specific receptors that recognize protease activities in allergens and the precise mechanisms that initiate type 2 allergic inflammation, particularly in the airways, remain unclear.

We demonstrate that detecting cysteine protease activity in HDM is crucial for initiating T helper type 2 (Th2) responses and type 2 immunity in the airways. This protease activity is recognized by protease-activated receptor 2 (PAR2) on a population of perivascular lung macrophages expressing Ly6G and the nuclear receptor Nr4a1/Nur77. These macrophages regulate the migration of conventional-migratory dendritic cells (mDC) to the mediastinal lymph nodes (mLN) by producing cysteinyl leukotrienes (CysLTs) for efficient T cell priming and expansion. Furthermore, inhibiting CysLT production reduces mDC migration and specifically weakens Th2 allergic responses without affecting other types of immune responses.

## Results

### PAR2 and protease sensing are required for Th2 cell responses to HDM

HDM is the most common cause of type 2 persistent allergic airway disease. In mice, intranasal (i.n.) HDM sensitization followed by i.n. HDM challenge (Fig. 1a) induced a robust accumulation of CD44[hi]CD4[+] T cells (Fig. 1c), IL-13/IL-5-producing Th2 cells (Fig. 1b, d), and eosinophils (Supplementary Fig. 1a) in the lung. To determine the role of different protease activities within HDM in promoting sensitization, HDM were either heat-inactivated (HI) or pretreated with irreversible inhibitors of cysteine (E-64) or serine (4-(2-aminoethyl)-benzenesulfonyl fluoride [AEBSF]) proteases to eliminate all or specific protease activity. These modified HDM preparations were then used to sensitize mice, followed by HDM challenge. Sensitization with HDM[HI] or HDM[E64] failed to induce accumulation of Th2 cells (Fig. 1b–d) and eosinophils (Supplementary Fig. 1a) in the lungs of challenged mice, whereas HDM[AEBSF] sensitization followed by HDM challenge induced similar Th2-driven inflammation as control HDM-sensitized and challenged mice (Fig. 1b–d, Supplementary Fig. 1a). These findings indicate that cysteine protease activity in HDM is the primary driver of the Th2 immune response. The papain-like cysteine protease Der p 1 is the most abundant and major allergen in HDM[14]. Sensitization and challenge with Der p 1 induced a robust accumulation of CD44[hi]CD4[+] T cells (Fig. 1f) and Th2 cells (Fig. 1e, g) in the lungs of mice, similar to HDM. This response depended on the cysteine protease activity of Der p 1, as E-64 pretreatment during Der p 1 sensitization prevented Th2 cell accumulation (Fig. 1e–g). Likewise, papain elicited a similar response (Fig. 1h–j), as expected from their shared enzymatic mechanism[15]. Overall, HDM, Der p 1, and papain induce Th2 responses through a common pathway that relies on cysteine protease activity.

Previous studies found that HDM can promote allergic inflammation due to LPS contamination[4]. Our HDM extract contained low endotoxin/LPS levels (<30 endotoxin units (EU)/mg). We compared its response to two additional standardized dried HDM extracts: one with high endotoxin levels (B82: ~8 × 10³ EU/mg) and another with low endotoxin levels (B91: 20 EU/mg). We found that our HDM extract and HDM[B91] potently induced Th2 responses (Supplementary Fig. 1b, c) and eosinophil accumulation (Supplementary Fig. 1f) in the lung, whereas the high-endotoxin HDM[B82] did not induce Th2-driven inflammation but instead promoted Th1 and Th17 cell accumulation in the lung (Supplementary Fig. b–f). These data suggest that LPS contamination suppresses type 2 immunity, as we have previously published[5–7]. To confirm this, we used Tlr4[−/−] mice, which lack the receptor for LPS. After sensitization and challenge with HDM, these mice generated robust Th2-driven responses, similar to WT controls (Supplementary Fig. 1g–k), demonstrating that our HDM does not rely on LPS or TLR4

activation to induce Th2 immune responses. Instead, cysteine protease activity in HDM is the primary driver of Th2 responses.

HDM sensitization induces the priming of allergen-specific CD4[+] T cells in the lung-draining mediastinal lymph node (mLN), which subsequently migrate to the lung after challenge[16]. To analyze differences in T-cell priming to HDM with or without cysteine protease activity, we transferred IL-4-GFP reporter (4get) OTII TCR-transgenic CD4[+] T cells, followed by sensitization with HDM or HDM[E-64] in the presence of OVA (Fig. 1k). HDM[E-64] was ineffective at inducing OTII cell expansion and GFP-IL-4 expression compared to HDM. Consequently, HDM[E-64] failed to induce the accumulation of IL-4[+] OTII cells in mLNs (Fig. 1l, m). Similar results were observed with HDM[HI] (Supplementary Fig. 1l, m). Likewise, sensitization with Der p 1 or papain in the presence of OVA induced robust OTII cell expansion and GFP-IL-4 expression, which was abrogated when allergens were pretreated with E-64 (Fig. 1n–p). These data suggest that cysteine protease activity in HDM, Der p 1, and papain is necessary for the expansion and Th2 lineage commitment of allergen-specific CD4[+] T cells.

PAR2 is a G protein-coupled receptor that becomes activated when its extracellular domain is cleaved by proteases, including cysteine proteases[2]. This activation has been implicated in allergic inflammation[2]. To investigate whether PAR2 is required to mediate responses to protease allergens, WT and Par2[−/−] mice were sensitized and challenged with papain (Fig. 1q). Th2 cells failed to accumulate in the lungs of Par2[−/−] mice compared to WT mice (Fig. 1r, s). We observed a similar response to HDM (Supplementary Fig. 1n, o). To test whether PAR2 expression was intrinsically needed in allergen-responsive CD4[+] T cells, we sensitized and challenged WT:Par2[−/−] mixed bone marrow (BM) chimeras with HDM (Supplementary Fig. 1p). We found no difference in the accumulation of naïve, activated CD44[hi] CD4[+] T cells, Th2 cells, or Der p 1:I-Ab-specific T cells in the lung or mLNs of the Par2[−/−] compartment compared to the WT compartment (Supplementary Fig. 1q). These results suggest that PAR2 essentially contributes to Th2 responses to HDM and other protease allergens but is not intrinsically required in responding T cells.

To analyze differences in T-cell priming in a PAR2-deficient environment, we transferred 4get.OT-II cells into WT and Par2[−/−] mice, sensitized the recipients, and analyzed the donor OT-II cells (Fig. 1t). OT-II cells did not accumulate in mLNs of Par2[−/−] mice compared with WT mice, although they still upregulated IL-4-GFP expression (Fig. 1u, v). The impaired expansion of OT-II cells resulted in a reduced number of IL-4[+] OT-II cells in the mLNs of Par2[−/−] mice (Fig. 1u, v). When these recipient mice were further challenged, we observed reduced accumulation of total OT-II and IL-4[+] OT-II cells in the lungs of Par2[−/−] mice (Fig. 1w–y). Similar results were obtained when recipient mice were treated with the PAR2 antagonist AZ3451 during allergen sensitization (Supplementary Fig. 1r, s). However, donor OT-II cell responses to i.n. immunization with LPS + OVA or to i.n. Influenza A virus strain PR8 engineered to express the OT-II OVA epitope (PR8-OTII) was comparable between Par2[−/−] and WT mice in both the mLNs and lungs (Supplementary Fig. 1t–x). These findings suggest that PAR2 is specifically required for Th2 responses to protease allergens.

### PAR2 and allergen protease sensing promote migration of mDCs

Since T cell priming was impaired in the absence of protease activity in allergens, we tested whether the capture of allergen by mDCs in the lung and their migration to the mLNs could also be defective. HDM and HDM[HI] were labeled with Alexa-647 prior to sensitization, and HDM-bearing, Alexa-647[+] cells in the lung and mLNs were analyzed 24 h later (Fig. 2a). No differences were found in HDM[+] cells in the lung, including total cells (Fig. 2b) or mDCs (both type 1 mDC1 and type 2 mDC2) (Fig. 2c), between HDM and HDM[HI]. These results suggest that protease inactivation of HDM does not affect the capacity of DC subsets to capture the allergen in the lung. However, we found reduced frequencies of HDM-bearing, Alexa-647[+] cells in the mLNs of mice treated with

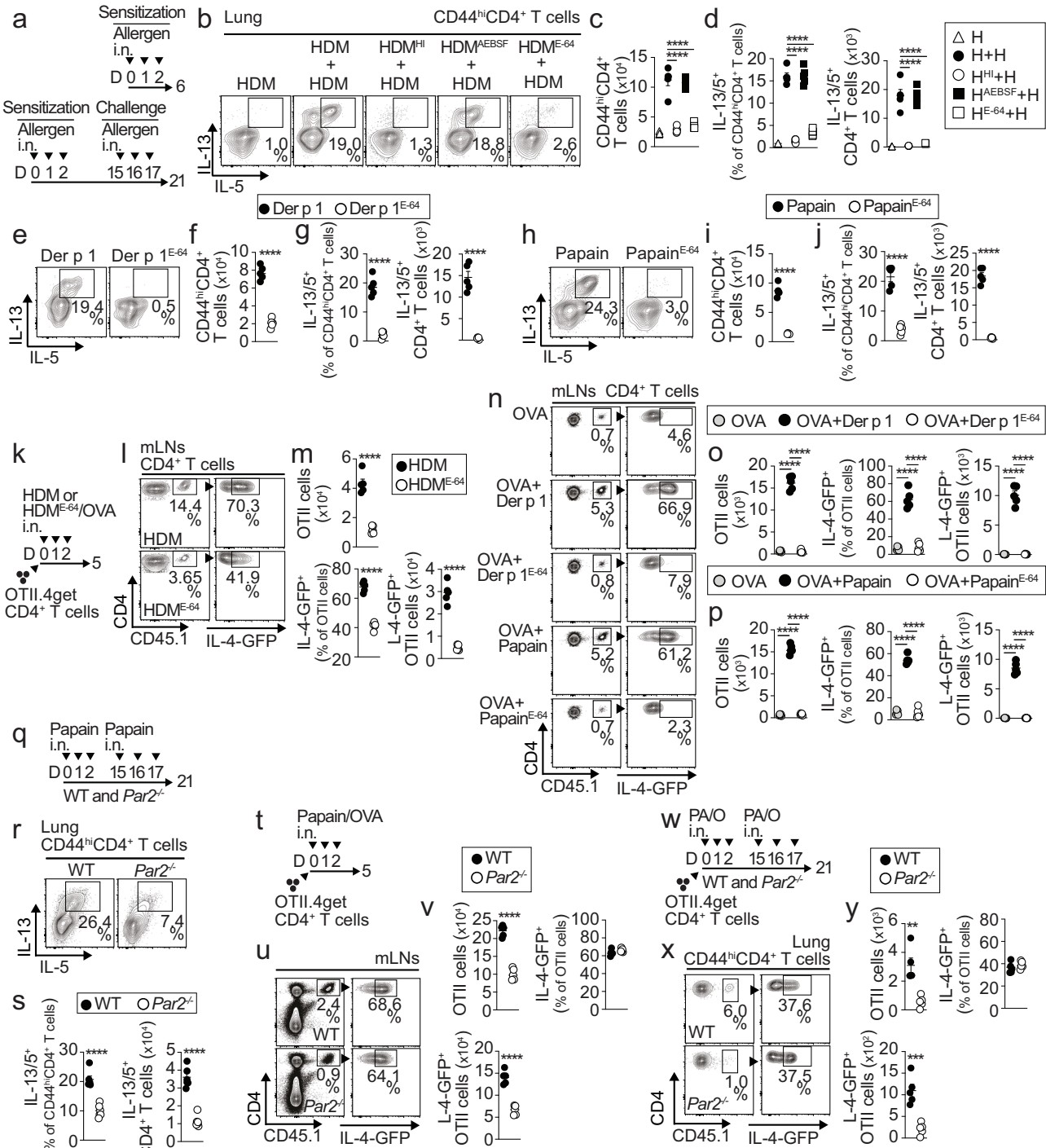

**Fig. 1 | PAR2-driven sensing of protease allergens is required to elicit Th2-driven airway inflammation. a** Schematic of the allergen sensitization and challenge protocols. **b–d** Representative plots of Th2 cells within gated lung CD44^hiCD4+ T cells after sensitization with normal or enzymatically inactivated HDM and challenge (**b**), as shown in (**a**), quantification of CD44^hiCD4+ T cells (**c**), and quantification of Th2 cells (**d**) (sensitization *n* = 3 mice, sensitization + challenge *n* = 5 mice per group). **e–j** Representative plots of Th2 cells within gated lung CD44^hiCD4+ T cells after sensitization with normal or enzymatically inactivated Der p 1 or papain and challenge (**e**, **h**), as shown in (**a**), quantification of CD44^hiCD4+ T cells (**f**, **i**), and Th2 cells (**g**, **j**) (*n* = 5 mice per group). **k–p** Schematic of OTII cell transfer and allergen sensitization (**k**), and representative plots (**l**, **n**) and quantification (**m**, **o**, **p**) of donor OTII and IL-4-GFP+ OTII cells from the mLN post-

sensitization (*n* = 5 mice). **q–s** Schematic of the papain sensitization and challenge (**q**), and representative plots (**r**), and quantification of Th2 cells in gated lung CD44^hiCD4+ T cells (**s**) (*n* = 6 mice per group). **t–y** Schematic of OTII cell transfer and papain sensitization (**t**) or sensitization and challenge (**w**) and representative plots (**u**, **x**) and quantification (**v**, **y**) of donor OTII and IL-4-GFP+ OTII cells in mLN (**u**, **v**) or lung (**x**, **y**) (*n* = 5 mice per group). (**c**, **d**, **o**, **p**) Statistical tests are one-way ANOVA with Dunnett's post hoc test. (**f**, **g**, **i**, **j**, **m**, **s**, **v**, **y**) Statistical tests are a two-tailed unpaired *t*-test. (**c**, **d**, **f**, **g**, **i**, **j**, **m**, **o**, **p**, **s**, **v**) ****p* < 0.0001. (**y**) ***p* = 0.0015; ****p* = 0.0005. Individual data points represent biological replicates, and bars show the mean ± SEM. Source data are provided in the Source Data file. Representative experiments of at least three independent repeats are shown.

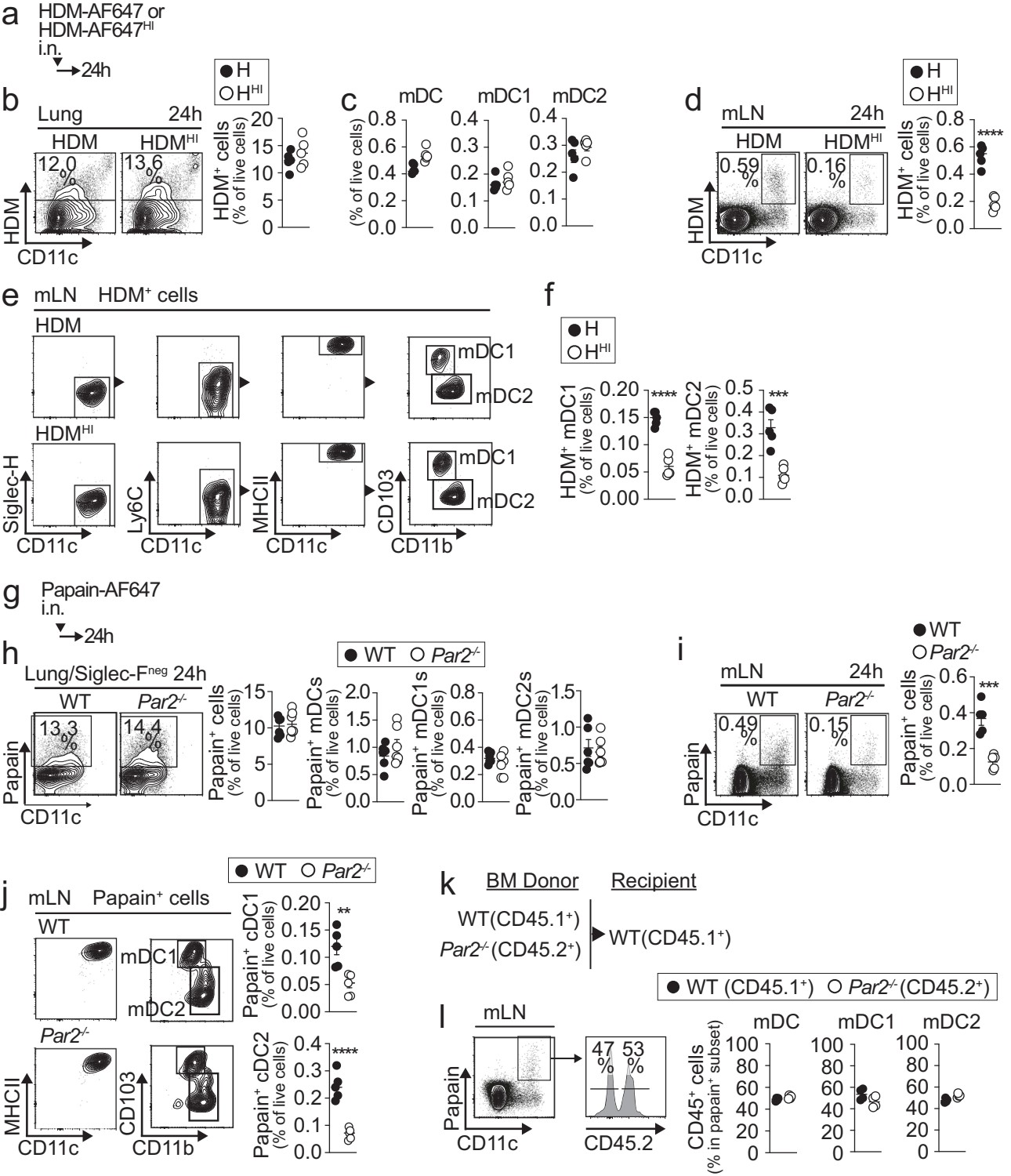

**Fig. 2 | PAR2-driven protease allergen sensing drives mDC migration to mLNs but is not mDC-intrinsic. a–f** Schematic of HDM sensitization (**a**) and representative plots and quantification of HDM⁺ (**b, d**) mDC, mDC1 and mDC2 (**c, e, f**) in lung (**b, c**) or mLN (**d–f**) (*n* = 5 mice per group). **g–j** Schematic of papain sensitization (**g**) and representative plots and quantification of papain⁺ total cells, mDC, mDC1, and mDC2 in lung or mLN (**h–j**) (**h**: WT *n* = 6 mice per group, *Par2*⁻/⁻ *n* = 7 mice per group; **i**: *n* = 5 mice per group). **k, l** Schematic of BM chimeric mice (**k**) and representative plots and quantification of papain⁺ mDC, mDC1, and mDC2 within CD45.1⁺ (WT) and CD45.2⁺ (*Par2*⁻/⁻) compartments in mLN (**l**) (*n* = 4 mice per group). All statistical tests are two-tailed, unpaired *t*-test. **d** *****p* < 0.0001. **f** *****p* < 0.0001; ****p* = 0.0008. **i** ****p* = 0.0004. **j** ***p* = 0.005 *****p* < 0.0001. Individual data points represent biological replicates, and bars show the mean ± SEM. Source data are provided in the Source Data file. Representative experiments of at least three were performed.

with HDM$^{HI}$ (Fig. 2d). Phenotypic analysis revealed that HDM$^+$ cells in mLNs corresponded exclusively to single-H$^-$Ly6C$^-$CD11c$^+$MHCII$^{hi}$ mDCs, including both CD103$^+$ mDC1 and CD11b$^+$ mDC2 (Fig. 2e). Both mDC1 and mDC2 were found in reduced frequencies in the mLNs of mice treated with HDM$^{HI}$ (Fig. 2f). These results suggest that protease activity in HDM is essential for promoting DC migration to the mLNs.

We next sensitized WT and $Par2^{-/-}$ mice with Alexa-647-labeled papain (Fig. 2g). No differences were found in papain$^+$ cells in the lung, including total cells or mDCs (mDC1 and mDC2) (Fig. 2h), between WT and $Par2^{-/-}$ mice 24 h after treatment. However, we observed reduced accumulation of papain$^+$ cells in the mLNs of $Par2^{-/-}$ mice (Fig. 2i), which included both mDC1 and mDC2 subsets (Fig. 2j). Similar results were obtained using HDM or when mice were treated with the PAR2 antagonist AZ3451 at the time of allergen exposure instead of using $Par2^{-/-}$ mice (Supplementary Fig. 2a–d). Since these data indicated that PAR2 is required for mDC migration to the mLNs in response to allergens, we tested whether PAR2 expression was intrinsically required in mDCs. Thus, we sensitized WT:$Par2^{-/-}$ mixed BM chimeras with Alexa-647-labeled papain (Fig. 2k). We found similar accumulation of papain$^+$ mDCs in both the $Par2^{-/-}$ and WT compartments. Overall, our data show that allergen protease activity and PAR2 are required for DC migration to the mLNs following allergen exposure, but PAR2 is not intrinsically required in mDCs.

## Atypical Ly6G$^+$ macrophages express and depend on PAR2 for lung accumulation following allergen exposure

To gain insight into which cells express PAR2 to regulate mDC and Th2 responses to allergens, we generated BM chimeric mice to determine whether PAR2 expression was required in the hematopoietic or structural compartment. Thus, irradiated hematopoietic-depleted WT and $Par2^{-/-}$ recipients were reconstituted with BM from either WT or $Par2^{-/-}$ mice to restore the hematopoietic compartment (Fig. 3a). Mice lacking $Par2$ in hematopoietic cells ($Par2^{-/-} > Par2^{-/-}$ and $Par2^{-/-} > $ WT) exhibited diminished lung Th2 cell responses compared to mice with WT hematopoietic cells (WT > WT and WT > $Par2^{-/-}$) following papain sensitization and challenge (Fig. 3b, c). Furthermore, mice with $Par2^{-/-}$ hematopoietic cells displayed defective migration of papain$^+$ mDCs to the mLNs 24 h after allergen exposure (Fig. 3d). These results suggest that PAR2 expression in the hematopoietic compartment is required for optimal mDC migration and Th2 responses to protease allergens. Next, to determine whether the required PAR2-expressing cells are host-resident or circulating, we surgically conjoined congenic WT and $Par2^{-/-}$ mice for 28 days to enable shared blood circulation, followed by exposure to Alexa-647-labeled papain. $Par2^{-/-}$ co-joined mice showed reduced migration of papain$^+$ mDCs to the mLNs 24 h after allergen exposure compared to WT co-joined mice (Fig. 3e). Although mDCs (WT and $Par2^{-/-}$) in the lungs of WT and $Par2^{-/-}$ mice took up papain similarly (Fig. 3f), both WT and $Par2^{-/-}$ mDCs in the mLNs of $Par2^{-/-}$ hosts contained significantly fewer papain$^+$ cells than mDCs in the mLNs of WT co-joined mice (Fig. 3g), suggesting that host-resident PAR2-expressing cells regulate mDC migration.

We next analyzed PAR2 expression in lung cells and found it associated with a distinct population characterized by the expression of monocyte-macrophage markers, including CX3CR1, F4/80, CD64, CD115, and Ly6C (Fig. 3h, i). Notably, this PAR2$^+$ population also expressed Ly6G, a marker typically associated with neutrophils (Fig. 3i), leading us to designate this unique population as Ly6G$^+$ macrophages (Ly6G$^+$ MΦ). Ly6G$^+$ MΦ homogeneously expressed PAR2, whereas other lung cells were negative (Supplementary Fig. 3a). Importantly, PAR2 surface expression on Ly6G$^+$ MΦ was markedly down-modulated 24 h after HDM or papain exposure compared with naïve controls, consistent with receptor activation and internalization following protease stimulation (Supplementary Fig. 3b). A most extensive phenotypic analysis of 24 h papain-treated mice showed that Ly6G$^+$ MΦ had a unique expression profile distinct from neutrophils, monocyte subsets,

or monocyte-derived DCs (moDCs). Ly6G$^+$ MΦ were characterized by high expression of molecules involved in adhesion and migration, such as CD11a, CD49d, CD43, CD66a, and CD97. Additionally, they highly expressed molecules involved in immune and homeostatic regulation, such as Treml4, Clec4a1, CD305, CD273, CD274, CD143, and CD36 (Supplementary Fig. 3b). Ly6G$^+$ MΦ also showed high staining for Ki-67 and EdU (5-ethynyl-2′-deoxyuridine), suggesting that these cells are highly proliferative (Supplementary Fig. 3c). To further investigate their proliferative dynamics, we performed an EdU pulse-chase experiment. We found that Ly6G$^+$ MΦ retained EdU labeling over time, although fluorescence intensity gradually declined, consistent with active proliferation of a stable population. In contrast, monocytes rapidly lost EdU signal, indicative of high turnover and replacement by newly generated cells (Supplementary Fig. 3d). Ly6G$^+$ MΦ exhibited high forward scatter (FSC) and side scatter (SSC) profiles, indicating that these cells are larger in size and internally granular, although less so than alveolar macrophages (Supplementary Fig. 3b). Morphology of Ly6G$^+$ MΦ was distinct from neutrophils, monocyte subsets and alveolar macrophages (Supplementary Fig. 3e). We found that Ly6G$^+$ MΦ in the lung uptake allergens and represented approximately 10–20% of the total cells that internalized HDM or papain (Fig. 3j–l). Other cells that uptaken allergens included alveolar macrophages, classical monocytes, moDCs, neutrophils, and mDCs (Fig. 3j–l). Ly6G$^+$ MΦ only constitute about 1% of the total cells in the lung (Fig. 3m, n), but their high representation among allergen$^+$ cells suggests a preferential ability of Ly6G$^+$ MΦ to take up allergens. Indeed, more than 60% of Ly6G$^+$ MΦ internalized HDM or papain, and along with alveolar macrophages, are the cell subsets with the most preferential ability to access and take up allergens in the lung (Supplementary Fig. 3f–h). Ly6G$^+$ MΦ remained of host origin after parabiotic surgery (Fig. 3o); however, they were radiosensitive and completely replaced by BM progenitors after irradiation depletion (Fig. 3p). In contrast, alveolar macrophages were radioresistant, with more than 70% remaining of host origin (Fig. 3p). Using $Ms4a3^{TdT}$ reporter mice, an established fate-mapping model for progeny of granulocyte-monocyte progenitors (GMPs)[17], we found that Ly6G$^+$ MΦ, as well as monocyte subsets, neutrophils, and eosinophils in the lung, all originate from GMPs (Fig. 3q) and were already detectable in the lung by postnatal day 7 (P7) (Supplementary Fig. 3i). In contrast, alveolar and interstitial macrophages were mostly not GMP-derived (Fig. 3q). We observed no defect in the generation of lung Ly6G$^+$ MΦ in $Ccr2^{-/-}$ mice (Fig. 3r), suggesting that Ly6G$^+$ MΦ may develop independently of CCR2$^+$ monocytes. However, after irradiation depletion, $Ccr2^{-/-}$ BM cells were defective in repopulating lung Ly6G$^+$ MΦ (Fig. 3s) and were completely unable to compete with WT BM for the generation of Ly6G$^+$ MΦ (Fig. 3t). These data suggest that CCR2$^+$ classical monocytes are precursors of Ly6G$^+$ MΦ when a niche is available, although compensatory pathways might be able to substitute for the absence of CCR2 during the homeostatic development of this population.

Ly6G$^+$ MΦ rapidly and transiently accumulated in the lungs of mice following allergen exposure (Fig. 3u). However, HDM$^+$ Ly6G$^+$ MΦ did not accumulate in the lungs of HDM$^{HI}$-treated mice compared to those treated with HDM (Fig. 3v). HDM$^+$ classical monocytes and moDCs were also reduced in the lungs of HDM$^{HI}$-treated mice (Fig. 3v), suggesting that the accumulation of allergen-bearing Ly6G$^+$ macrophages, classical monocytes, and moDCs depends on protease activity in HDM. Similarly, total and papain$^+$ Ly6G$^+$ MΦ did not accumulate in the lungs of treated $Par2^{-/-}$ mice compared to WT mice (Fig. 3w, Supplementary Fig. 3j). moDCs also did not differentiate (Fig. 3w), indicating that PAR2 is required for the accumulation of allergen-bearing Ly6G$^+$ MΦ and moDCs. To test whether PAR2 expression was intrinsically required in these cells, we sensitized WT:$Par2^{-/-}$ mixed BM chimeras with labeled papain (Fig. 3x). We found that papain$^+$ $Par2^{-/-}$ Ly6G$^+$ MΦ severely failed to accumulate in the lung, and only WT Ly6G$^+$ MΦ were able to accumulate in the lung (Fig. 3y). In contrast, other papain$^+$ lung cells showed similar accumulation in both the $Par2^{-/-}$ and

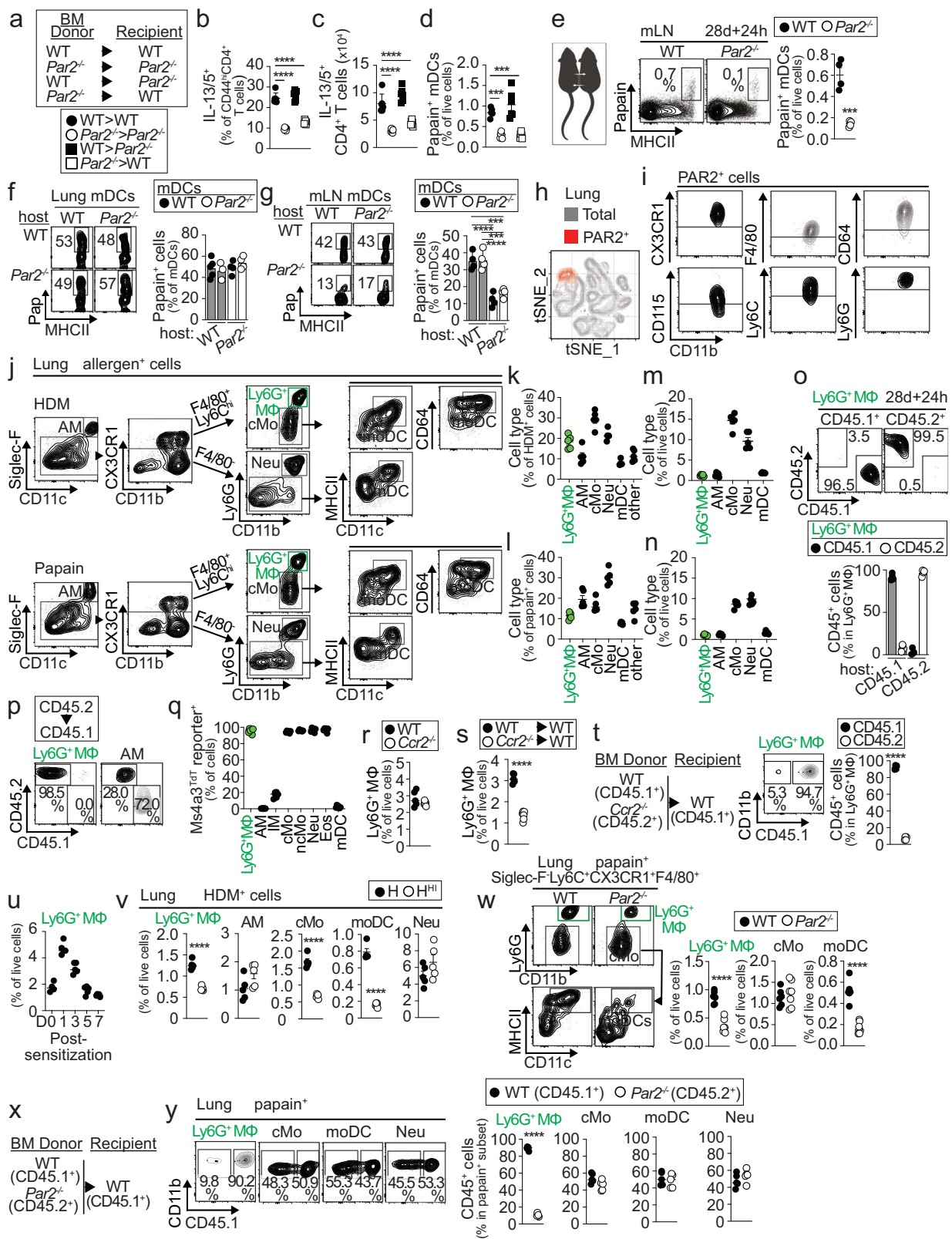

WT compartments (Fig. 3y). Thus, Ly6G⁺ MΦ require PAR2 for lung accumulation after allergen exposure. Overall, Ly6G⁺ MΦ are a population of macrophages with a unique expression profile, high proliferative capacity, and superior ability to take up inhaled allergens. Ly6G⁺ MΦ are radiosensitive and derive from GMP progenitors and CCR2⁺ monocytes. Ly6G⁺ MΦ accumulate in the lung after treatment with protease allergens and require PAR2 for this process.

## Ly6G⁺ MΦ represent a transcriptionally distinct perivascular MΦ subset in the lung

To understand Ly6G⁺ MΦ in relation to other cells that encounter and capture allergens in the lung—beyond population-level analysis—we transcriptionally profiled individual papain⁺ cells from the lung 24 h post-treatment using scRNA-seq. Papain⁺ cells clustered into 17 transcriptional groups, one of which (cluster 4) expressed a signature

**Fig. 3 | PAR2 expression by Ly6G⁺ MΦ is required for their lung accumulation following allergen exposure. a–d** Schematic of BM chimeric mice (**a**), lung Th2 cell quantification after papain sensitization and challenge (**b, c**), and papain⁺ mDC quantification in mLNs 24 h post-exposure (**d**) (*n* = 6 mice per group). **e–g** Papain⁺ mDC quantification in mLNs 24 h post-exposure in parabiotic mice joined for 28 days (**e**) and papain⁺ mDC quantification based on their origin (donor mouse) and residence (host mouse) in the lung (**f**) and mLNs (**g**) (*n* = 4 mice per group). **h, i** Representative plot showing PAR2⁺ cell clusters in the lung (**h**) and phenotype of PAR2⁺ cluster cells (**i**). **j–n** Flow gating of allergen⁺ cells from the lung 24 h post-exposure (**j**) and quantification of allergen⁺ (**k, l**) and total (**m, n**) cells (*n* = 5 mice per group). **o, p** 24 h post-allergen exposure assessment of MΦ populations in the lung using congenic markers CD45.1 and CD45.2 after 28 days of parabiosis (**o**) (*n* = 4 mice per group) or in BM chimeras (**p**). **q** Expression of *Ms4a3*-tdTomato in the indicated lung cells (*n* = 6 mice per group). **r–t** 24 h post-allergen exposure assessment of Ly6G⁺ MΦ in the lung of WT and *Ccr2⁻/⁻* mice (**r**) (*n* = 5 mice per group), WT and *Ccr2⁻/⁻* BM chimeras (**s**) (*n* = 5–6 mice per group), and WT:*Ccr2⁻/⁻* competitive BM chimeras (**t**) (*n* = 4 mice per group). **u** Kinetic quantification of Ly6G⁺ MΦ in the lung after allergen exposure (*n* = 5 mice per group). **v** HDM⁺ Ly6G⁺ MΦ quantification in the lung 24 h post-exposure to native or heat-inactivated HDM (*n* = 5 mice per group). **w–y** Flow gating and quantification of indicated papain⁺ cells in the lung 24 h post-exposure in WT and *Par2⁻/⁻* mice (**w**) (*n* = 6 mice per group) and WT:*Par2⁻/⁻* competitive BM chimeras (**x, y**) (*n* = 5 mice per group). Abbreviations: AM, alveolar macrophages; Eos, eosinophils; IM, interstitial MΦ; cMo, classical monocytes; ncMo, non-classical monocytes; Neu, neutrophils. **b–d** Statistical tests are one-way ANOVA with Dunnett's post hoc test. **b, c** ****$p < 0.0001$. **d** ***$p = 0.0002$ WT > WT vs *Par2⁻/⁻* > *Par2⁻/⁻* and *Par2⁻/⁻* > WT. **g** Statistical tests are two-way ANOVA with Tukey's post hoc test. ****$p < 0.0001$ WT/WT vs WT/*Par2⁻/⁻*; ***$p = 0.0004$ WT/WT vs *Par2⁻/⁻*/*Par2⁻/⁻*; ****$p < 0.0001$ *Par2⁻/⁻*/WT vs WT/*Par2⁻/⁻*;***$p = 0.0003$ *Par2⁻/⁻*/WT vs *Par2⁻/⁻*/*Par2⁻/⁻*. (**e, s, t, v, w, y**) Statistical tests are a two-tailed unpaired *t*-test. **e** ***$p = 0.0006$. (**s, t, v, w, y**) ****$p < 0.0001$. Individual data points represent biological replicates, and bars show the mean ± SEM. Source data are provided in the Source Data file. Representative experiments of at least three were performed. Panel (**e**) was created with BioRender. Leon, B. (2026). https://BioRender.com/gup034j.

associated with Ly6G⁺ MΦ (Supplementary Fig. 4a, b). Other clusters were categorized based on their transcriptional signatures into alveolar macrophages (clusters 2, 14), neutrophils (clusters 0, 1, 6, 15), classical/CCR2⁺ monocytes (clusters 5, 7), moDCs (clusters 3, 11), and conventional DCs, including type 1 (cluster 8), type 2, (cluster 13) and migratory (cluster 10) DC phenotypes. Additionally, we identified smaller clusters representing endothelial cells, NK cells, and T cells (Supplementary Fig. 4a, b). Signature genes of Ly6G⁺ MΦ included markers highly expressed in these cells: *Csf1r* (CD115), *Cx3cr1*, *Fcgr4* (CD16.2), *Itgal* (CD11a), *Itga4* (CD49d), *Spn* (CD43), *Clec4a1*, *Treml4*, *Cd36*, *Ace* (CD143), and *Lair1* (CD305) (Supplementary Fig. 4b, c). Other notable genes included *Nr4a1*, *Bcl2*, *Adgre4*, *Pparg*, *Eno3*, *Tgfbr3*, and *Cd300e* (Supplementary Fig. 4c). *Ly6g* transcripts were not detected in any cell in the scRNA-seq dataset. We isolated the Ly6G⁺ MΦ cluster along with other monocyte-related clusters (classical monocytes and moDCs) for further analysis. In this refined analysis, Ly6G⁺ MΦ segregated into two clusters (cluster 2, 3) (Fig. 4a). Since classical monocytes are precursors to both Ly6G⁺ MΦ (Fig. 3s, t) and moDCs[6], we explored their differentiation pathways using pseudotime trajectory analysis. This revealed two distinct trajectories, one leading to cluster 3 and subsequently to cluster 2 within the Ly6G⁺ MΦ population, suggesting classical monocytes differentiate into Ly6G⁺ MΦ through two developmental states. The second trajectory led to moDCs, indicating an alternative differentiation pathway. These findings highlight the plasticity of classical monocytes, capable of giving rise to both Ly6G⁺ MΦ and moDCs via separate pathways (Supplementary Fig. 4d). We then performed Gene Set Enrichment Analysis (GSEA) using signatures of monocyte subsets (Supplementary Data 1[18].). This analysis revealed that Ly6G⁺ MΦ clusters were enriched for a non-classical monocyte signature, whereas classical monocyte and moDC clusters were enriched for the classical monocyte signature (Fig. 4b). These findings suggest that Ly6G⁺ MΦ may differentiate through a pathway involving a non-classical monocyte transition. Gene ontology (GO) analysis showed that Ly6G⁺ MΦ clusters were highly associated with adhesion, proliferation and DNA repair, differentiation, and activation processes (Fig. 4c, Supplementary Data 2). Notably, cluster 2 displayed more GO terms related to activation than cluster 3, suggesting it may represent a more activated state of Ly6G⁺ MΦ. To further characterize Ly6G⁺ MΦ relative to non-classical monocytes (ncMo), we performed an integrated scRNA-seq analysis of isolated populations. The integrated UMAP revealed that the two cell types clustered separately, demonstrating distinct transcriptional identities (Supplementary Fig. 5a, b). ncMo were predominantly localized to clusters 0 and 1, whereas Ly6G⁺ MΦ distributed across clusters 2.0, 2.1, 3, 4, and 5 (Supplementary Fig. 5c). Despite their distinct clustering, both populations shared expression of a core set of genes including *Csf1r*, *Cx3cr1*, *Nr4a1*, *Bcl2*, *Pparg*, *Tgfbr3*, *Il10ra*, *Klf2*, *Klf4*, *Clec4a1*, *Cebpb*, *Itga4*, *Itgal*, *Spn*, *Cd9*, *Cd36*, *Lair1*, *Treml4*, *Fcgr4*, and *Ace*. However, Ly6G⁺ MΦ uniquely expressed higher levels of genes associated with effector and inflammatory programs, such as *Arg2*, *Dhrs9*, *Dusp1*, *Hdc*, *Alox5*, *Ptgs2*, *Il1r2*, *Il1rn*, *Lmnb1*, and *Ly6g* (Supplementary Fig. 5d). To explore the relationship between ncMo and Ly6G⁺ MΦ, we performed pseudotime trajectory analysis. From ncMo, two branching developmental pathways were observed: Branch 1 was enriched for pathways associated with cell migration, activation, adhesion, angiogenesis, vasculature regulation, and cytokine/chemokine production, suggesting a differentiation program leading to immune activation/regulation and tissue remodeling. Branch 2 was enriched for pathways related to translation, ribosome biogenesis, oxidative phosphorylation (OXPHOS), nucleoside/nucleotide synthesis, protein folding, apoptosis, and p53 stress signaling, indicating a metabolic and stress-response program (Supplementary Fig. 5e, f, Supplementary Data 3). Cell cycle analysis revealed that differentiation of Ly6G⁺ MΦ along Branch 1 was accompanied by an increased proportion of G2/M-phase cells, consistent with active proliferation during this trajectory (Supplementary Fig. 5g). These data indicate that Ly6G⁺ MΦ represent a specialized macrophage lineage with a developmental relationship to ncMo.

We next performed RNA-seq to analyze gene expression changes during Ly6G⁺ MΦ development by comparing the transcriptomes of Ly6G⁺ MΦ and their precursors, CCR2⁺ classical BM monocytes. Approximately 1500 genes were differentially upregulated by at least a two-fold change. These included signature genes of Ly6G⁺ MΦ (Fig. 4d) that were previously identified at the transcript level in our scRNA-seq analysis (Supplementary Fig. 4b, c, Fig. 5d) and at the protein level by flow cytometry (Supplementary Fig. 3b). Ly6G⁺ MΦ were enriched for a non-classical monocyte signature and negatively enriched for the classical monocyte signature (Fig. 4e, Supplementary Data 1), consistent with our scRNA-seq analyses (Fig. 4b, Supplementary Fig. 5d). Ingenuity Pathway Analysis (IPA) identified biological functions in Ly6G⁺ MΦ that were similar to those observed in our scRNA-seq analysis, including adhesion, migration, movement, proliferation, and inflammatory response (Fig. 4f, Supplementary Data 4). IPA canonical pathways were associated with degranulation, alternative macrophage activation via IL-4/13, tissue remodeling and fibrosis, and tumor microenvironment (Fig. 4g, Supplementary Data 4). We then performed GSEA using an atlas of human macrophage subset signatures previously identified across various tissues during prenatal development[19]. Ly6G⁺ MΦ showed strong enrichment for a human perivascular angiogenic macrophage subset (Fig. 4h and Supplementary Data 5), which is predominantly found in the lung and skin[19]. Since these data suggested that Ly6G⁺ MΦ might represent a perivascular population in the lung, we conducted

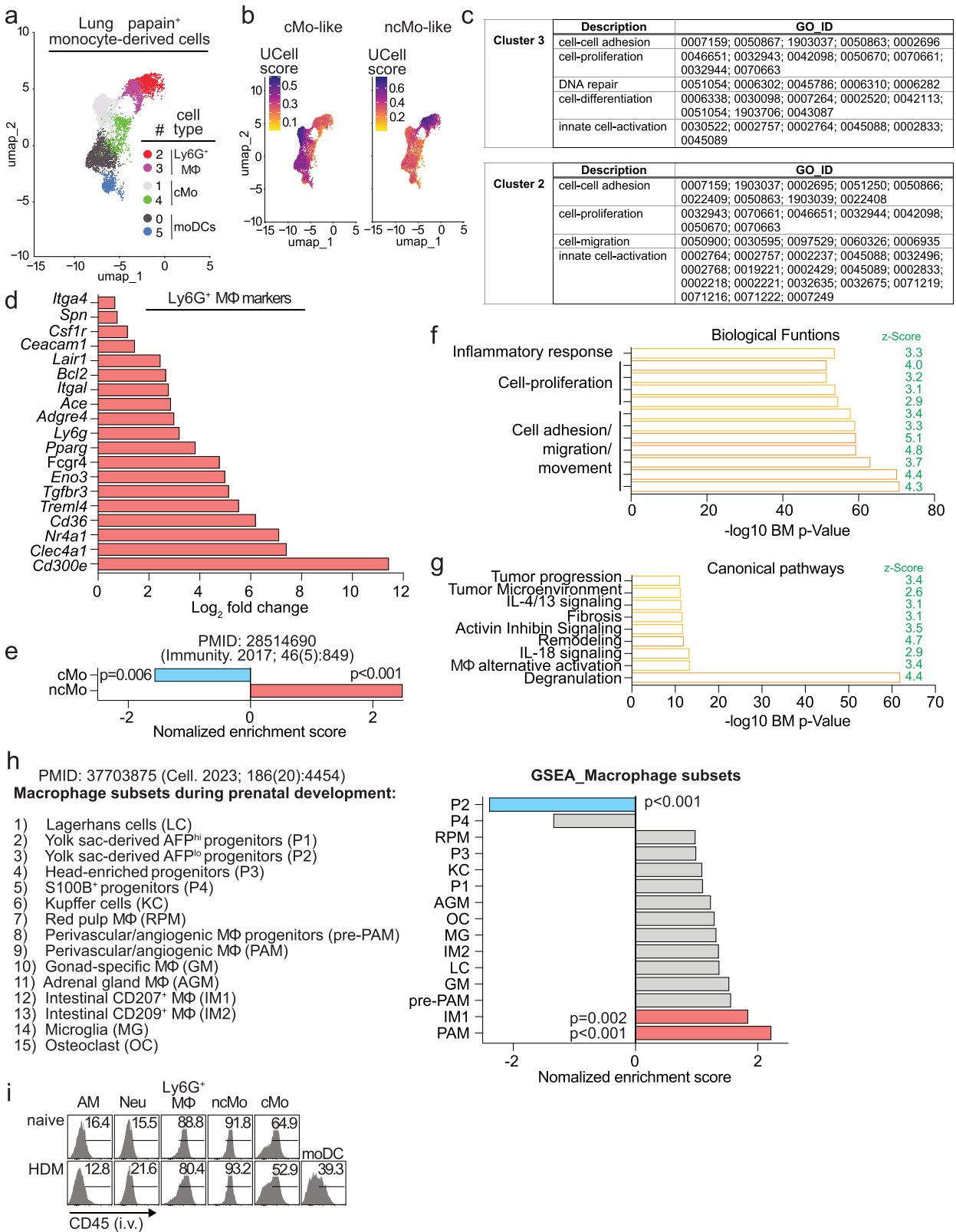

short-term intravascular labeling experiments. Mice were intravenously (i.v.) injected with labeled CD45 prior to analysis to stain circulating cells and those in perivascular locations accessible from the bloodstream. We found that Ly6G+ MΦ were highly stained with i.v. CD45 in both naïve and allergen-treated mice (Fig. 4i). Additionally, monocyte subsets were labeled. However, unlike monocytes, Ly6G+ MΦ were not detected in circulation (Supplementary Fig. 5h), suggesting perivascular

localization in the lung. In contrast, moDCs, neutrophils, and alveolar macrophages were protected from labeling (Fig. 4i), indicating that they resided within lung tissue. Overall, our findings indicate that Ly6G+ MΦ constitute a population of perivascular macrophages characterized by a non-classical monocyte signature, with features of increased cell adhesion, migration, proliferation, alternative macrophage activation, and tissue remodeling.

**Fig. 4 | Ly6G⁺ MΦ constitute a unique population of perivascular macrophages in the lung. a, b** Single-cell RNA-seq Uniform Manifold Approximation and Projection (UMAP) plots of papain⁺ monocyte-derived cells, showing annotated clusters (**a**) and enrichment for gene signatures of classical and non-classical monocytes (**b**). **c** Table showing top GO pathways in Ly6G⁺ MΦ clusters as in (**a**). **d** Fold change in the Ly6G⁺ MΦ signature gene core from RNA-seq data comparing Ly6G⁺ MΦ isolated from naïve lung to precursor CCR2⁺ BM monocytes (pooled-mouse samples; $n = 3$ per group). **e** Enrichment of gene signatures of classical and non-classical monocytes in Ly6G⁺ MΦ versus precursor CCR2⁺ BM monocytes. Statistical significance was determined using the GSEA permutation test; normalized enrichment score and false discovery rate (FDR q-value) are shown. **f, g** Top

activated biological functions (**f**) and canonical pathways (**g**) in Ly6G⁺ MΦ versus precursor CCR2⁺ BM monocytes based on IPA using a right-tailed Fisher's exact test (reported as -$\log_{10}$(p-value)). Predicted activation is indicated by the IPA activation z-score. **h** Enrichment of macrophage subset gene signatures across tissues in Ly6G⁺ MΦ compared to precursor CCR2⁺ BM monocytes. Statistical significance was determined using the GSEA permutation test; normalized enrichment score and FDR q-value are shown. **i** Representative plots of CD45⁺ cells in the indicated lung populations, 5 min after i.v. CD45 injection. Abbreviations: AM, alveolar macrophages; cMo, classical monocytes; ncMo, non-classical monocytes; Neu, neutrophils.

## Ly6G⁺ MΦ intrinsically depend on the transcription factor Nr4a1/Nur77 for their function in controlling the migration of mDCs and the initiation of Th2 responses

Since one of the top identified signature genes of Ly6G⁺ MΦ in our scRNA-seq and RNA-seq analyses was *Nr4a1* (Fig. 4d, Supplementary Fig. 4b, c and Supplementary Fig. 5d), we used NR4A1-GFP reporter mice to study *Nr4a1* expression upon allergen exposure. We found that *Nr4a1* was not expressed in mLNs 24 h after allergen exposure but was present in the lung, where its expression was specifically restricted to Ly6G⁺ MΦ and ncMo (Supplementary Fig. 6a, b). Among lung cells that took up allergens, only Ly6G⁺ MΦ expressed *Nr4a1* (Fig. 5a). Since *Nr4a1* is a specific marker for Ly6G⁺ MΦ, we used NR4A1-GFP reporter mice to assess the location of Ly6G⁺ MΦ in the lung. We found that Nr4a1⁺ Ly6G⁻ CX3CR1⁺ MΦ were primarily localized along the CD31⁺ capillary network within the perivascular-interstitial compartment of the alveolar parenchyma (Supplementary Fig. 6c). We next used *Nr4a1* enhancer domain E2-deficient (*se_2⁻/⁻*) mice, which lack *Nr4a1* expression specifically in the monocyte lineage[20], to analyze the role of *Nr4a1* in Ly6G⁺ MΦ. We found that *se_2⁻/⁻* mice exhibited defective accumulation of Ly6G⁺ MΦ and ncMo in the lung 24 h following allergen exposure (Fig. 5b). To test whether *Nr4a1* was intrinsically required in allergen-bearing Ly6G⁺ MΦ, we sensitized WT:*se_2⁻/⁻* mixed BM chimeras with labeled papain (Fig. 5c). We found that papain⁺ *se_2⁻/⁻* Ly6G⁺ MΦ failed to accumulate in the lung, with WT Ly6G⁺ MΦ accounting for >80% of the accumulation (Fig. 5c). In contrast, other papain⁺ lung cells showed similar accumulation in both the *se_2⁻/⁻* and WT compartments (Fig. 5c). Thus, Ly6G⁺ MΦ require the transcription factor *Nr4a1* (Nur77) for lung accumulation following allergen exposure. Ly6G⁺ MΦ significantly increased Ki-67 expression in papain-treated WT mice, indicating cell proliferation following allergen-driven activation (Supplementary Fig. 6d). However, in papain-treated *se_2⁻/⁻* mice, Ly6G⁺ MΦ failed to upregulate Ki-67 (Supplementary Fig. 6d), suggesting a defect in cell cycle entry, which may contribute to the impaired accumulation of Ly6G⁺ MΦ in the lung in these mice.

Since *se_2⁻/⁻* mice have defects in Ly6G⁺ MΦ, we assessed Th2 responses in these mice. We found that Th2 cells failed to accumulate in the lungs of *se_2⁻/⁻* mice compared to WT mice after sensitization and challenge with papain (Fig. 5d). Similar results were observed with HDM (Supplementary Fig. 6e). However, we found that *Nr4a1se_2* expression was not intrinsically required in allergen-responsive CD4⁺ T cells (Supplementary Fig. 6f). Additionally, we observed that *se_2⁻/⁻* mice had impaired expansion of donor OT-II cells in the mLNs following allergen sensitization (Fig. 5e) and reduced accumulation of total OT-II and IL-4⁺ OT-II cells in the lungs following challenge (Fig. 5f). We next evaluated mDC migration in *se_2⁻/⁻* mice and observed reduced accumulation of papain⁺ mDC1 and mDC2 cells in the mLNs of *se_2⁻/⁻* mice 24 h after treatment, compared to WT mice (Fig. 5g). No differences were found in papain⁺ cells in the lungs, including mDCs (Supplementary Fig. 6g), indicating defective migration of mDCs to the mLNs in *se_2⁻/⁻* mice. However, in WT:*se_2⁻/⁻* mixed BM chimeras treated with labeled papain, we found similar accumulation of papain⁺ mDCs in both the *se_2⁻/⁻* and WT mLN compartments (Fig. 5h), suggesting that *Nr4a1se_2* expression is not intrinsically required in mDCs for

migration to the mLNs. Overall, our data suggest that *Nr4a1se_2* expression is intrinsically required in Ly6G⁺ MΦ to control mDC migration to the mLNs and drive allergen-specific Th2 responses. To confirm this idea, we generated *Par2⁻/⁻:se_2⁻/⁻* mixed BM chimeras (50% *s/P⁻/⁻*), reasoning that Ly6G⁺ MΦ, uniquely dependent on both genes, would be selectively impaired in these mice compared with control WT:*se_2⁻/⁻* and WT:*Par2⁻/⁻* BM chimeras (Fig. 5i). 24 h after papain treatment, we found a defect in allergen-driven lung accumulation of Ly6G⁺ MΦ in 50% *s/P⁻/⁻* chimeras compared with control chimeras, as expected (Fig. 5j). Other lung cell populations were unaffected (Fig. 5j). Accompanying the Ly6G⁺ MΦ defect, we observed defective migration of mDCs to the mLNs (Fig. 5k), defective expansion of donor OTII cells in the mLNs after sensitization (Fig. 5l), and defective accumulation of Th2 cells (Fig. 5m), donor OTII cells (Fig. 5n), and eosinophils (Fig. 5m) in the lung after challenge (Fig. 5). This suggests that defective activation and accumulation of Ly6G⁺ MΦ after allergen exposure impair mDC migration to the mLNs and the development of Th2 responses and type 2 inflammation. Immunofluorescence of lung sections showed that in naïve lungs, CCR7⁺ DCs were sparsely distributed throughout the parenchyma (Supplementary Fig. 6h) and were often found in close proximity to Nr4a1⁺ Ly6G⁺ MΦ near CD31⁺ capillaries within the alveolar-airway parenchyma (Supplementary Fig. 6i), whereas after papain treatment, CCR7⁺ DCs clustered around LYVE-1⁺ lymphatics (Supplementary Fig. 6h). These findings raise the possibility that activation of Ly6G⁺ MΦ may help guide DCs toward afferent lymphatics to promote their migration to the mLNs.

## Ly6G⁺ MΦ promote mDCs migration by producing cysteinyl leukotrienes (CysLTs)

CCR7 directs mDC migration to mLNs[21]. To test whether CCR7 is required for this migration in response to papain, we generated WT:*Ccr7⁻/⁻* BM chimeras (Fig. 6a). In the lung, papain⁺ WT and *Ccr7⁻/⁻* mDCs were equivalently represented 24 h after allergen administration (Fig. 6b). However, only papain⁺ WT mDCs were detected in mLNs (Fig. 6b), suggesting that CCR7 expression in mDCs is essential for their migration to mLNs in response to papain. Interestingly, although mDCs upregulated CCR7 expression following common Th1/Th17 stimuli, such as LPS treatment or PR8 influenza infection, we observed a reduction in CCR7 expression in papain-treated mice compared with naïve mice (Fig. 6c). This prompted us to question how mDCs are still able to migrate increasingly to mLNs in response to papain, even when CCR7 expression is reduced. Since Ly6G⁺ MΦ are required for this enhanced migration, we examined the differentially expressed genes in our Ly6G⁺ MΦ RNA-seq database. We found specific genes related to the eicosanoid signaling network to be upregulated, particularly *Alox5*, *Alox5ap*, and *Ltc4s* (Supplementary Fig. 7a, b), which encode critical enzymes involved in the synthesis of CysLTs, including LTC4, LTD4, and LTE4 (Supplementary Fig. 7a). Flow cytometry confirmed that Ly6G⁺ MΦ express high levels of ALOX5 in naïve mice compared to other cell populations in the lung (Supplementary Fig. 7c). Additionally, we analyzed changes in eicosanoid pathway genes in Ly6G⁺ MΦ from naïve and papain-treated mice and found significant upregulation of *Ltc4s* in Ly6G⁺ MΦ from papain-treated

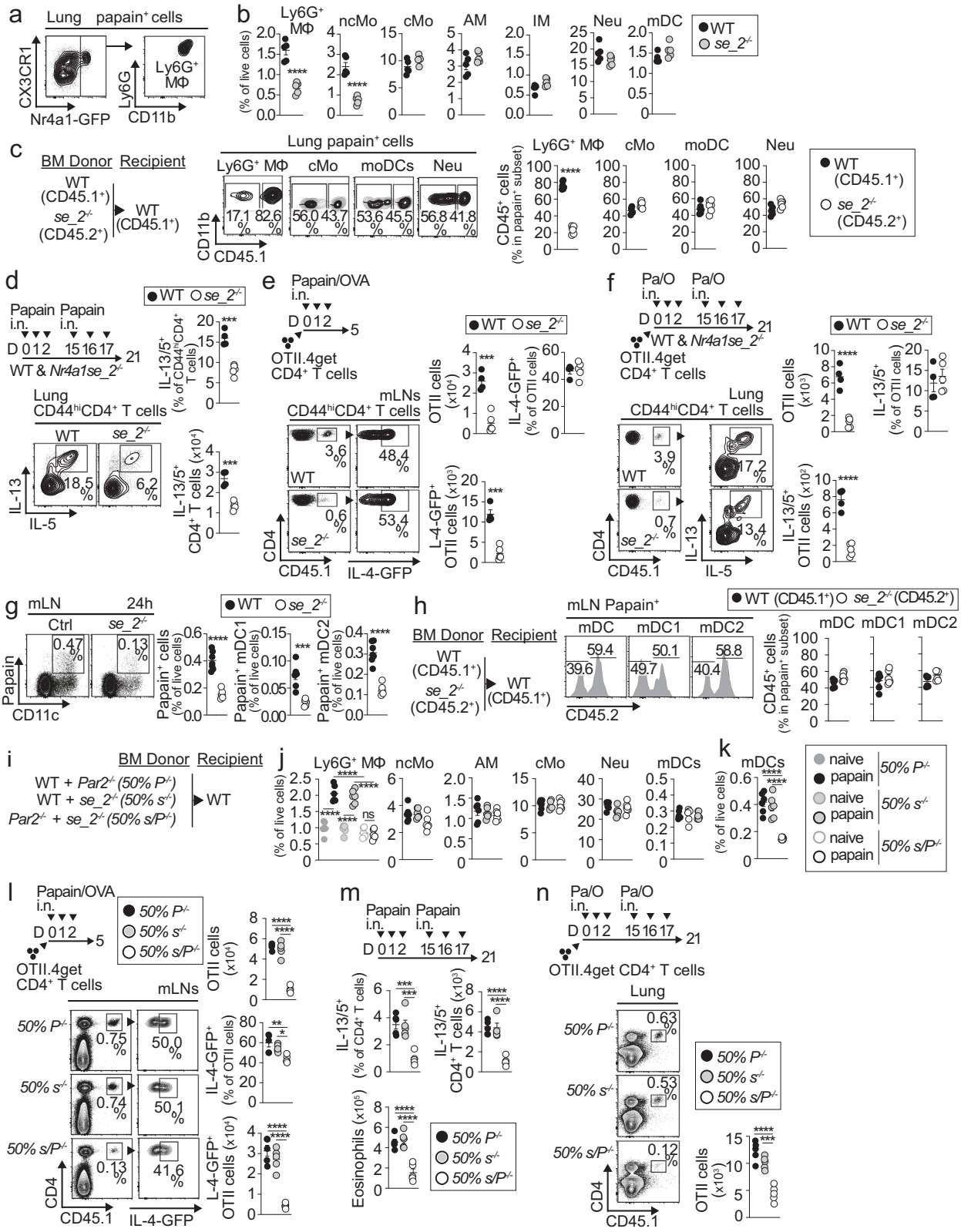

mice (Fig. 6d). Overall, these results suggest that Ly6G⁺ MΦ have the capability to produce CysLTs, and this activity may be enhanced by exposure to allergens. To directly assess the capacity of Ly6G⁺ MΦ to produce CysLTs, we purified Ly6G⁺ MΦ from the lungs and stimulated them ex vivo with papain. WT Ly6G⁺ MΦ robustly produced LTC4, whereas PAR2⁻/⁻ Ly6G⁺ MΦ displayed impaired production (Supplementary Fig. 7d), indicating that papain-induced

PAR2 signaling promotes LTC4 production in these cells. In line with this, we observed that in vivo exposure to papain caused a transient increase in LTC4 levels in the lungs (Fig. 6e). However, this response was not present in *Par2⁻/⁻* and *Nr4a1⁻/⁻* mice (Fig. 6f), which have defective Ly6G⁺ MΦ.

To test whether CysLTs are required for mDC migration to mLNs following papain treatment, we generated BM chimeras using WT and

**Fig. 5 | Nr4a1/Nur77 expression in Ly6G⁺ MΦ is essential for Th2-driven airway inflammation to allergens. a** Representative plots of *Nr4a1*-GFP⁺ cells within papain⁺ cells in the lung. **b** Quantification of indicated cells in the lung 24 h papain post-exposure in WT and *se_2⁻/⁻* mice (*n* = 5 mice per group). **c** Schematic, representative plots, and quantification of indicated papain⁺ cells in the lung 24 h post-exposure in WT:*se_2⁻/⁻* competitive BM chimeras (*n* = 6 mice per group). **d** Schematic of allergen exposure in WT and *se_2⁻/⁻* mice, representative plots of Th2 cells in gated lung CD44⁺CD4⁺ T cells, and Th2 cell quantification (WT *n* = 4 mice and *se_2⁻/⁻ n* = 5 mice per group). **e, f** Schematic of OTII cell transfer and papain sensitization (**e**) or sensitization and challenge (**f**), representative plots of donor OTII cells in mLN or lung, and quantification of total, IL-4-GFP⁺, and IL-13/5⁺ OTII cells in WT and *se_2⁻/⁻* mice (*n* = 5 mice per group). **g** Representative plots and quantification of papain⁺ total cells, mDC1, and mDC2 in mLN of WT and *se_2⁻/⁻* mice (WT *n* = 7 mice and *se_2⁻/⁻ n* = 5 mice per group). **h** Schematic, representative plots, and quantification of papain⁺ mDC, mDC1, and mDC2 within CD45.1⁺ (WT) and CD45.2⁺ (*se_2⁻/⁻*) compartments in mLN (*n* = 6 mice per group). **i, n** Schematic of BM chimeras (**i**), quantification of indicated cells in the lung (**j**) (*n* = 6 mice per group) and mLNs (**k**) (*n* = 6 mice per group) in naïve and 24 h post-papain exposure, schematic, representative plots of donor OTII cells in mLN, and quantification of total and IL-4-GFP⁺ OTII cells (**l**) (*n* = 5 mice per group), schematic and quantification of Th2 cells and eosinophils in the lung (**m**) (*n* = 5 mice per group), and schematic, representative plots, and quantification of donor OTII cells in the lung (**n**) (*n* = 5 mice per group). Abbreviations: AM, alveolar macrophages; IM, interstitial MΦ; cMo, classical monocytes; ncMo, non-classical monocytes; Neu, neutrophils. **b–g** Statistical tests are a two-tailed unpaired *t*-test. (**b, c, f, g**) ****p < 0.0001. **d** ***p = 0.0001; ***p = 0.0002. **e** ***p = 0.0003; ***p = 0.0001. **g** ***p = 0.0004. **j–n** Statistical tests are one-way ANOVA with Tukey's post hoc test. **j–n** ****p < 0.0001. **l** **p = 0.0033; *p = 0.0349. **m** ***p = 0.0004 *P⁺* vs *s/P⁺*; ***p = 0.0006 *s⁻/⁻* vs *s/P⁺*. **n** ***p = 0.0002. Individual data points represent biological replicates, and bars show the mean ± SEM. Source data are provided in the Source Data file. Representative experiments of at least three were performed.

*Alox5⁻/⁻* BM donor cells and sensitized them with papain. As expected, *Alox5⁻/⁻* BM chimeras exhibited reduced LTC4 levels in the lung after papain treatment (Fig. 6g). Although allergen capture in the lung was similar between *Alox5⁻/⁻* and WT mice, we observed reduced accumulation of papain⁺ mDCs in mLNs of *Alox5⁻/⁻* mice compared to WT mice (Fig. 6h), suggesting defective migration from the lung to the mLN. This defect was rescued by LTC4 inoculation at the time of allergen exposure (Fig. 6h). These findings indicate that LTC4 production is essential for mDC migration to mLNs following allergen exposure. In addition, we show that both Ly6G⁺ MΦ-deficient Par2⁻/⁻ and se_2⁻/⁻ mice, which had defective LTC4 production and impaired mDC migration to the mLN; this latter defect could be rescued by LTC4 or LTD4 inoculation at the time of allergen exposure, but not by LTB4 inoculation (Fig. 6i, Supplementary 7e–g). This suggests a link between Ly6G⁺ MΦ and CysLT production in the control of mDC migration to allergens. To confirm this idea, we generated *se_2⁻/⁻*:*Alox5⁻/⁻* mixed BM chimeras (*s/A⁻/⁻*) to selectively target *Alox5* deficiency to Ly6G⁺ MΦ. Since *se_2⁻/⁻* BM cells are unable to efficiently reconstitute Ly6G⁺ MΦ in the lung, most Ly6G⁺ MΦ in these mice are expected to be derived from *Alox5⁻/⁻* BM cells and therefore expected to lack the capacity to produce CysLTs. We compared these mice to control WT:*se_2⁻/⁻* (WT/*s⁻/⁻*) and WT:*Alox5⁻/⁻* (WT/*A⁻/⁻*) BM chimeras (Fig. 6j). We confirmed that both experimental *s/A⁻/⁻* and control mice had normal numbers of Ly6G⁺ MΦ after papain sensitization (Fig. 6k). Additionally, no differences were observed in other lung cells (Supplementary Fig. 7h). As expected, Ly6G⁺ MΦ in experimental *s/A⁻/⁻* chimeras showed defective expression of ALOX5 compared to control chimeras. (Fig. 6l, m). No differences in ALOX5 expression were found in other lung cells (Supplementary Fig. 7i). We found *s/A⁻/⁻* chimeras were unable to produce detectable levels of LTC4 in the lung following papain exposure (Fig. 6n), suggesting Ly6G⁺ MΦ are a major source of CysLTs in the lung post-allergen exposure. We also found reduced accumulation of papain⁺ mDCs in mLNs of *s/A⁻/⁻* mice compared to control mice (Fig. 6o) despite similar allergen capture in the lung (Supplementary Fig. 7j), suggesting defective migration from the lung to the mLN. Furthermore, this migration defect was rescued by LTC4 inoculation at the time of allergen exposure (Fig. 6o, Supplementary Fig. 7j). These results indicate that Ly6G⁺ MΦ regulate mDC migration to mLN after allergen exposure by producing CysLTs. Finally, to address how CysLTs may regulate CCR7-dependent migration of mDCs toward their ligands, CCL19 and CCL21, we assessed mDCs from mice treated with either papain or LPS in in vitro chemotaxis assays. mDCs from papain-treated mice exhibited poor migration toward CCL21 compared to those from LPS-treated mice (Fig. 6p, q). However, the addition of LTC4 or LTD4 enhanced migration to CCL21, while no effect was observed for CCL19 (Fig. 6p). These findings suggest that LTC4 and LTD4 specifically enhance CCR7-dependent mDC migration toward CCL21.

## Inhibition of CysLT production blocks mDC migration and prevents Th2 immune responses to allergens

Blocking the action of CysLTs on CysLT receptor 1 (CysLT1R) using drug antagonists is a common treatment for asthma and allergic rhinitis[22]; however, no current drug blocks CysLT2R and CysLT3R, which have distinct but complementary roles to CysLT1R in mediating CysLT signaling[23]. As a potentially more effective approach, we tested whether broadly inhibiting CysLT production could prevent Th2 immunity. First, we tested whether treatment of mice with the LTC4 synthase (LTC4S) inhibitor AZD9898 at the time of allergen exposure could impact mDC migration to the mLNs. AZD9898 treatment strongly reduced mDC migration to the mLNs in papain-treated mice (Fig. 7a, b, Supplementary Fig. 7k) but had no effect in mice treated with papain + LPS (Fig. 7a, c), suggesting that LTC4S blockade specifically inhibits mDC migration to allergens. Consistent with this, we found that a single AZD9898 treatment at the onset of allergen sensitization (Fig. 7d) blocked the expansion of donor OTII cells in the mLNs of papain-sensitized mice (Fig. 7e, f) but not in mice additionally treated with LPS (Fig. 7e, g). Finally, a single AZD9898 treatment during allergen sensitization effectively suppressed Th2 responses in the lung following allergen challenge (Fig. 7h, i). These data suggest that LTC4S inhibition during allergen exposure can effectively prevent the development of Th2 cell responses. To determine whether LTC4S inhibition could also prevent the development of type 2 inflammation in sensitized mice, we administered AZD9898 at the beginning of the challenge phase rather than during sensitization (Fig. 7j, l). Strikingly, this later treatment produced similar effects, including reduced mDC migration and re-expansion of donor OTII cells in the mLNs (Fig. 7j, k), and suppressed Th2 responses (Fig. 7l, m) and eosinophilia (Fig. 7n) in the lung. These findings indicate that LTC4S inhibition is effective at both the sensitization and challenge phases, highlighting its therapeutic potential for allergic airway disease.

In conclusion, we have identified a unique perivascular/pericapillary Ly6G⁺ macrophage population that highly expresses PAR2 and *Nr4a1*/Nur77 and responds to cysteine protease activity within aeroallergens. Activation of these Ly6G⁺ macrophages is essential for driving a full Th2 allergic response, as they regulate mDC migration to mLNs by producing LTC4, which synergizes with CCR7 on mDCs to enhance migration toward CCL21, thereby facilitating T cell priming and expansion. Importantly, targeting the LTC4S-driven CysLT biosynthesis pathway effectively and specifically reduces mDC migration to mLNs following allergen exposure and represents a promising strategy for preventing and treating Th2 cell-driven allergic inflammation.

## Discussion

The innate immune system recognizes foreign antigens by detecting specific, conserved patterns commonly found in pathogens[1]. However,

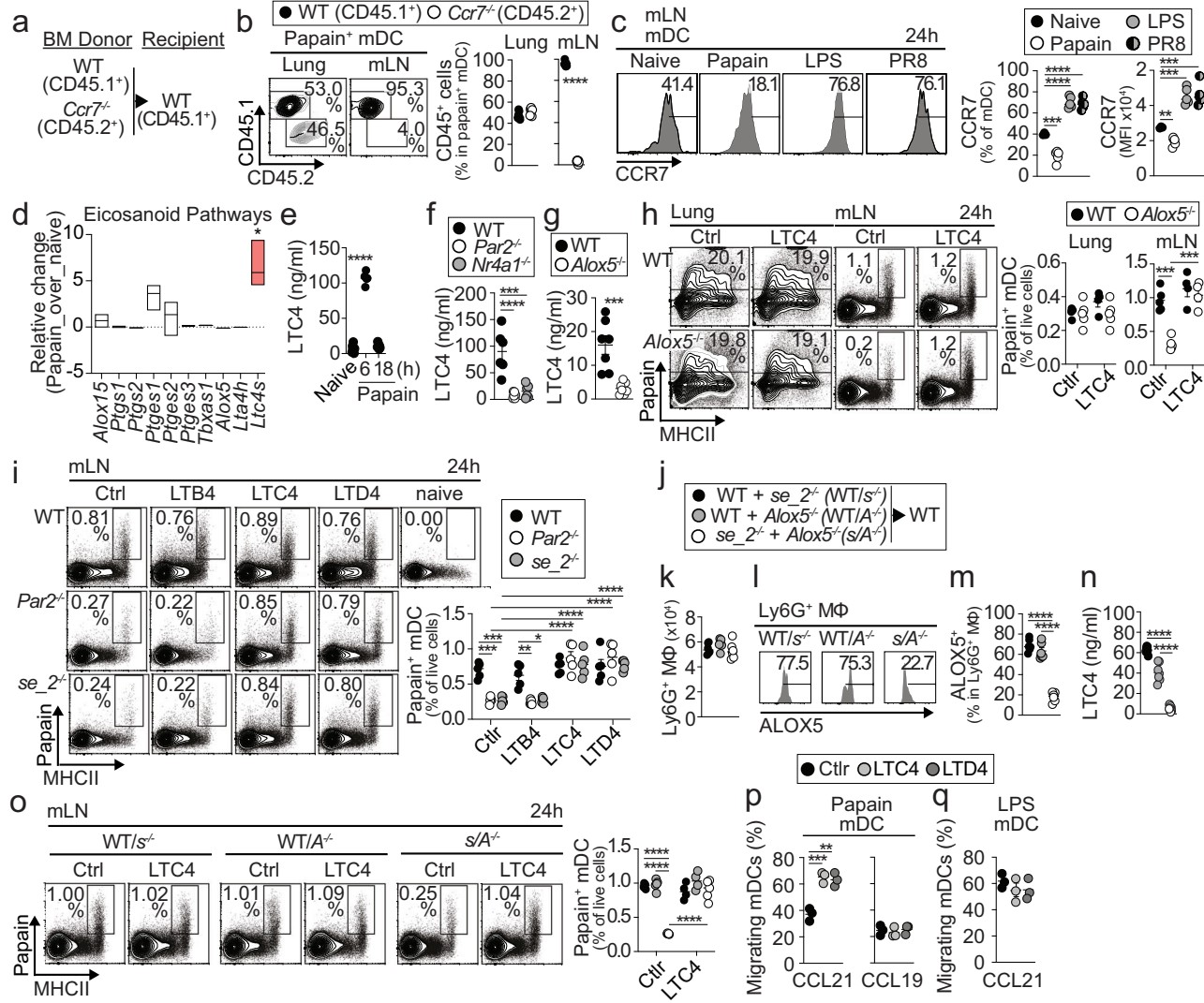

**Fig. 6 | Ly6G+ MΦ produce CysLTs upon allergen exposure, driving mDC migration to mLNs. a, b** Schematic of BM chimeras and representative plots and quantification of papain+ mDC within CD45.1+ (WT) and CD45.2+ (*Ccr7*−/−) compartments in lung and mLN (**b**) (*n* = 7 mice per group). **c** Representative plots and quantification of CCR7 expression on mDCs from mLNs in mice i.n. treated with papain, LPS, or PR8 (naive *n* = 3 mice, treated *n* = 5 mice per group). **d** Change in expression of key genes involved in eicosanoid pathways in Ly6G+ MΦ from naïve and papain-treated mice (pooled-mouse samples; *n* = 4 per group). **e, f** LTC4 lung levels in indicated mice after papain treatment (**e**: naive *n* = 7 mice, papain 6 h *n* = 5 mice, papain 6 h *n* = 7 mice per group; **f**: *n* = 6 mice per group). **g, h** LTC4 lung levels and representative plots and quantification of papain+ total cells and mDCs in lung and mLN of papain-treated WT and *Alox5*−/− BM chimeric mice, with and without LTC4 co-treatment (**g**: *n* = 7 mice per group; **h**: Ctrl *n* = 5 mice, LTC4 *n* = 4–5 mice per group). **i** Representative plots and quantification of papain+ mDCs in mLN of WT, *Par2*−/−, and *se_2*−/−, with and without co-treatment with the indicated LTs (*n* = 5 mice per group). **j–o** Schematic of BM chimeras (**j**), quantification of lung Ly6G+ MΦ after papain treatment (**k**) (*n* = 6 mice per group), representative plots and quantification of ALOX5 expression in Ly6G+ MΦ (**l, m**) (*n* = 6 mice per group), LTC4 lung levels (**n**) (*n* = 7 mice per group), and representative plots and

quantification of papain+ mDCs in mLN of indicated mice, with and without LTC4 co-treatment (**o**) (Ctrl *n* = 5 mice, LTC4 *n* = 4–5 mice per group). **p, q** Quantification of migration of mDCs from papain (**p**)- and LPS (**q**)-treated mice in transwell chemotaxis assays, using CCL21 or CCL19 as chemoattractants, with or without LTC4 or LTD4 treatment (*n* = 3 mice per group). **b, g** Statistical tests are a two-tailed unpaired *t*-test. **b** ****p < 0.0001. **g** ***p = 0.0007. (**c, e, f, h, m–p**) Statistical tests are one-way ANOVA with Tukey's post hoc test. **c** left panel: ***p = 0.002 naïve vs papain; ****p < 0.0001 and right panel: **p = 0.0014 naïve vs papain; ***p = 0.0002 naïve vs LPS; ***p = 0.0001 naïve vs PR8. **e** ****p < 0.0001. **f** ****p < 0.0001; ***p = 0.0002. **h** ***p = 0.0002 Ctlr WT vs Ctlr *Alox5*−/−; ***p = 0.0001 Ctlr *Alox5*−/− vs LTC4 *Alox5*−/−. **m** ****p < 0.0001. **n** ****p < 0.0001. **o** ****p < 0.0001. **p** ***p = 0.0008 Ctlr vs LTC4; **p = 0.0012 Ctlr vs LTD4. **d** Statistical test is a two-sided one-sample *t*-test, followed by Benjamini–Hochberg FDR correction. *p < 0.0374. **i** Statistical tests are two-way ANOVA with Tukey's post hoc test. ****p < 0.0001; ***p = 0.0006 Ctlr WT vs Ctlr *Par2*−/−; ***p = 0.0006 Ctlr WT vs Ctlr *se_2*−/−; **p = 0.0033 LTB4 WT vs LTB4 *Par2*−/−; *p = 0.0101 LTB4 WT vs LTB4 *se_2*−/−. Individual data points represent biological replicates, and bars show the mean ± SEM. Box plots display the median and the minimum and maximum values. Source data are provided in the Source Data file. Representative experiments of at least three were performed.

existing evidence suggests that this mechanism may not apply to allergen detection. Instead, our study supports the notion that allergens are sensed by the immune system based on their ability to disrupt tissue homeostasis through potentially harmful activities. Specifically, we demonstrated that cysteine protease activity in HDM, associated with the papain-like major allergen Der p 1, is the major trigger for innate cell activation and subsequent development of Th2 and type 2

immunity. This is in line with other studies highlighting the role of protease activity in allergens linked to allergic pathogenicity[2,10–13,24–26]. In this context, the detection of abnormal tissue protease activity, particularly cysteine protease activity, may have evolved as part of a mechanism for sensing tissue damage and triggering a response to combat or repair it. However, not all abnormal protease activity poses a threat, and this host-driven protease-sensing mechanism can

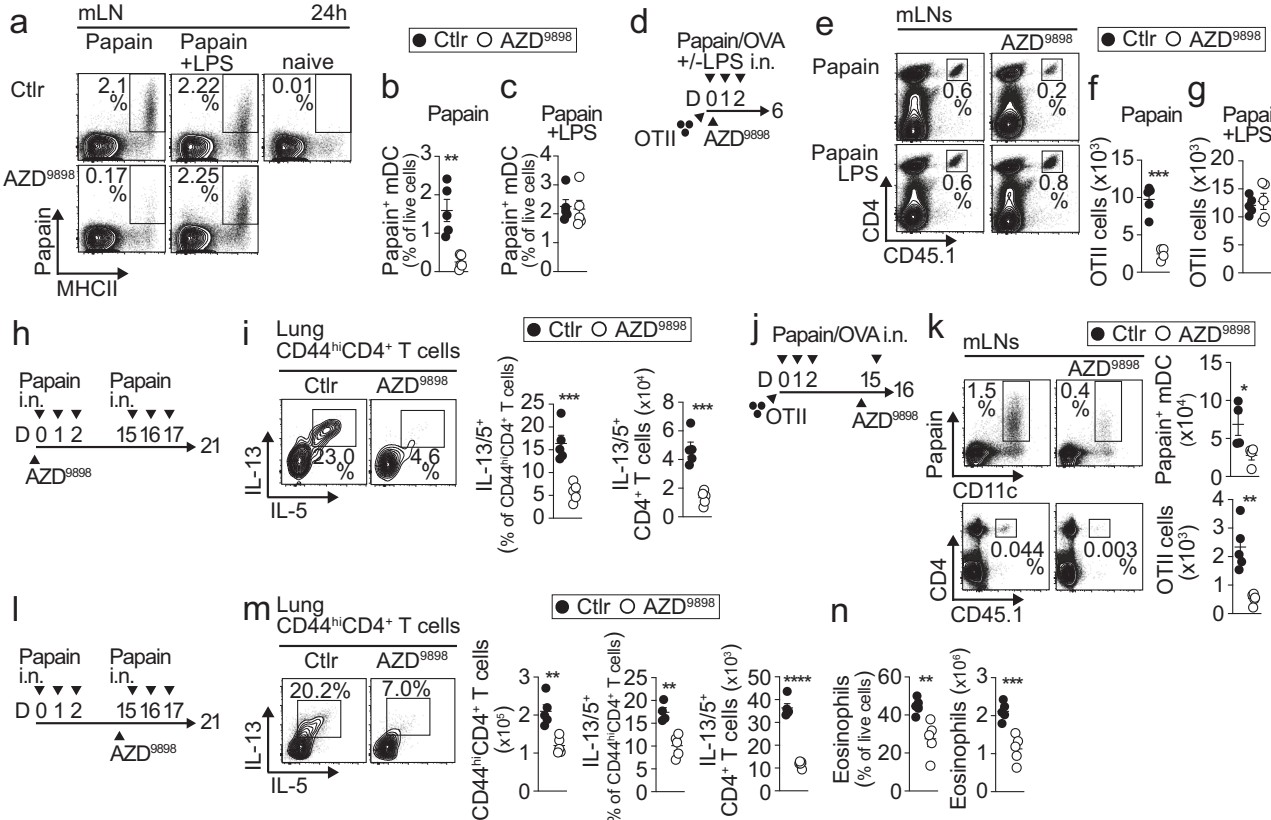

**Fig. 7 | Blockade of CysLT synthesis specifically and effectively blocks Th2 responses to allergens. a–c** Representative plots and quantification of papain+ mDCs in mLN of mice co-treated or not with LPS and/or the LTC4S inhibitor AZD9898 ($n = 5$ mice per group). **d–g** Schematic of OTII cell transfer and treatments (**d**), representative plots of donor OTII cells in mLN (**e**), and quantification of OTII cells (**f, g**) ($n = 5$ mice per group). **h, i** Schematic of treatments (**h**) and representative plots and quantification of Th2 cells in lung (**i**) ($n = 5$ mice per group). **j, k** Schematic of OTII cell transfer and treatments (**j**), and representative plots and quantification of papain+ mDCs ($n = 4$ mice per group) and donor OTII cells ($n = 5$

mice per group) in mLN (**k**). **l–n** Schematic of treatments (**l**), representative plots and quantification of Th2 cells (**m**), and eosinophils (**n**) in the lung ($n = 5$ mice per group). (**b, f, i, k, m, n**) Statistical tests are a two-tailed unpaired *t*-test. **b** **$p = 0.0021$. **f** ***$p = 0.0002$. **i** ***$p = 0.0007$ left; ***$p = 0.0002$ right. **k** *$p = 0.0406$; **$p = 0.0014$. **m** **$p = 0.0017$ left; **$p = 0.0018$ center; ****$p < 0.0001$. **n** **$p = 0.0049$; ***$p = 0.0008$. Individual data points represent biological replicates, and bars show the mean ± SEM. Source data are provided in the Source Data file. Representative experiments of two were performed.

sometimes be detrimental, as seen in the case of exposure to protease allergens.

In our study, we have identified a population of lung macrophages that play a crucial role in initiating a protease-sensing immune mechanism. These macrophages, which we have named Ly6G+ MΦ, form a distinct population of perivascular macrophages located in the alveolar interstitium, occupying a strategic position between the alveoli and capillaries. This location grants them preferential access to inhaled allergens, enabling efficient uptake. Ly6G+ MΦ arise from GMP progenitors and are maintained locally as a long-lived, self-renewing population; however, they can be replenished by CCR2+ monocytes through a non-classical monocyte pathway only following depletion. Our data indicate that Ly6G+ MΦ sense protease activity in allergens via the PAR2 receptor. In response to this detection, they get activated and accumulate, a process that involves, at least in part, their proliferation. This mechanism is regulated by the transcription factor *Nr4a1*/Nur77, which is highly expressed in this macrophage population. Previous studies suggested a role for the PAR2 receptor in driving allergic responses[2,27,28], but the specific immune mechanisms involved remained unclear. Our study identifies Ly6G+ MΦ as the key PAR2+ cells responsible for PAR2-mediated sensing of protease allergens and initiating Th2-driven allergic inflammation. Importantly, *Nr4a1*/Nur77 expression and activation are also required for this process in Ly6G+ macrophages, although the mechanistic relationship between PAR2 and Nr4a1/Nur77 in Ly6G+ macrophage activation remains to be

defined. Additionally, transcriptomic analyses of Ly6G+ MΦ revealed pathways and functions associated with increased cell adhesion, migration, proliferation, IL-13/IL-4-driven macrophage activation, and tissue remodeling, suggesting that there may be additional roles for Ly6G+ MΦ yet to be explored.

We found that Ly6G+ MΦ are different from other classical populations of macrophages in the lung, which include alveolar macrophages located in the alveolar space and interstitial macrophages located in the lung parenchyma[29]. Other studies have described atypical macrophage populations with features partly resembling the Ly6G+ MΦ identified in our study, including macrophages displaying a bi-lobed or segmented nuclear morphology[30] and Ly6G expression[31] in the lung during fibrosis or alveolar regeneration after injury. These populations may be related to the one we describe; however, their transcriptomic signatures, developmental trajectories, and anatomical localization differ from ours, suggesting that they may represent distinct macrophage subsets or reflect different activation states arising from specific experimental contexts.

Our study emphasizes the crucial role of allergen-driven activation of Ly6G+ MΦ in regulating the migration of mDCs to the mLNs, which is essential for initiating allergen-specific T cell responses. The migration of mDCs to mLNs is primarily mediated by the chemokine receptor CCR7[21,32,33], which is upregulated in response to normal homeostatic maturation[34–36] or inflammatory conditions[21,37–40]. We found that pathogen-driven stimuli, such as LPS-induced inflammation

or influenza infection, strongly induced CCR7 expression in mDCs in the lung. However, exposure to allergens had the opposite effect, inducing CCR7 downregulation in mDCs. This finding is consistent with our previous studies showing downregulation of CCR7 during type 2 responses induced by intestinal helminths[41]. This CCR7 downregulation is important for allowing the positioning of mDCs at the T-B cell boundary in mLNs, away from the T cell area, and facilitating encounters with CD4+ T cells in these regions, which is crucial for initiating the priming of Th2 responses[8,16,21,33,41–44]. However, a paradox arises regarding how mDCs can still exhibit robust arrival to mLNs despite the downregulation of CCR7 following allergen exposure. Our study identifies cysLT produced by Ly6G+ MΦ as a key signal supporting CCR7-dependent migration of mDCs in the context of allergen exposure. Additionally, we found that this supporting function of cysLT on mDC migration is critical following allergen stimulation but does not play a role following other inflammatory conditions that strongly upregulate CCR7. As such, our data show that inhibiting cysLT production by a drug inhibitor specifically suppresses mDC migration to allergens and subsequent type 2 inflammation, highlighting a targeted therapeutic approach for type 2 immunity. CysLT1R antagonists like montelukast, zafirlukast, and pranlukast, when used consistently, reduce asthma symptoms[22]. While their precise mechanism remains unclear, previous studies have shown that CysLT1R signaling in innate lymphoid cells (ILC2) can promote type 2 inflammation[45–48]; thus, blocking CysLT1R signaling may help mitigate this response. Our findings further suggest that CysLT1R antagonists may help alleviate the development of an ongoing Th2 response, contributing to the suppression of type 2 inflammation. However, these drugs only partially block cysLT activity, as CysLT2R and CysLT3R can compensate for CysLT1R. Thus, pan-inhibition of cysLT synthesis may offer a more effective alternative.

## Methods

### Animals

All animal procedures were conducted in accordance with institutional guidelines and approved by the UAB and NIAID Institutional Animal Care and Use Committee (IACUC). Mice were housed under specific pathogen-free conditions. Unless otherwise stated, mice were 6–12 weeks old, maintained on a C57BL/6 J background, and both sexes were used in balanced numbers. Animals were euthanized using $CO_2$ asphyxiation followed by cervical dislocation in accordance with approved protocols. The mice strains used in these experiments include: C57BL/6 J (B6), B6.SJL-Ptprc[a] Pepc[b]/BoyJ (CD45.1+ B6 congenic), C57BL/6-Tg(TcraTcrb)425Cbn/J (OTII), B6.129-Il4[tm1Lky]/J (B6.4get IL-4-GFP reporter), B6.Cg-F2rl1[tm1Mslb]/J (Par2−/−), C57BL/6-Rr39[em1Ched]/J (Nr4a1se_2/se_2), B6;129S2-Nr4a1[tm1Jmi]/J (Nur77−/−), B6.129S4-Ccr2[tm1Ifc]/J (Ccr2−/−), C57BL/6-Tg(Nr4a1-EGFP/cre) 820Khog/J (Nur77GFP), B6(Cg)-Tlr4[tm1.2Karp]/J (Tlr4−/−), B6;129S2-Alox5[tm1Fun]/J (Alox5−/−), C57BL/6J-Ms4a3[em2(cre)Fgnx]/J (Ms4a3cre), and B6.Cg-Gt(ROSA)26Sor[tm9(CAG-tdTomato)Hze]/J (Ai9). Nr4a1se_2/se_2 were developed and kindly provided by Dr. Catherine C. Hedrick. All other mice were originally obtained from Jackson Laboratory and bred onsite. To produce BM chimeras, recipient mice were irradiated with an 800 Rad split dose from a high-energy X-ray source and reconstituted with $10^7$ total BM cells. Mice were analyzed 7–8 weeks later. For parabiosis experiments, sex- and weight-matched mice were cohoused for 20 days before surgery. The flanks of the partnered mice were shaved 2–3 days prior to the operation. Longitudinal incisions were made from the forelimb to the hindlimb, and sutures were used to join the flanks of the mice. Mice were analyzed 4 weeks post-surgery.

### Allergen exposure, adoptive transfer and treatments

Mice were i.n. administered with the following allergens: 100 μg of HDM (Dermatophagoides pteronyssinus) extract (Greer Laboratories),

10 μg of Papain (Sigma-Aldrich), or 10 μg of Der p 1 (Prospec) for 1–3 days, followed by an i.n. challenge with 100 μg of HDM, 10 μg of Papain, or 10 μg of Der p 1 for 3 days, 2 weeks apart. In some experiments, allergens were co-administered with 5 μg endotoxin-free OVA (EndoFit OVA, InvivoGen). In certain experiments, allergens were heat-inactivated (100 °C for 45 min) or treated with protease inhibitors: E-64 (100 nM, 37 °C for 30 min) or AEBSF (1 mM, 37 °C for 30 min) (Sigma-Aldrich). Some allergens were labeled with Alexa Fluor 647 (Invitrogen) before administration. Additional experimental conditions included i.n. administration of PAR2 antagonist AZ3451 (30 μg, MedChem Express), LPS from Escherichia coli O111:B4 (5 μg, Sigma-Aldrich), or 2 nM of Leukotriene B4, C4, or D4 (Cayman Chemical Company) at the time of initial sensitization. For influenza virus infection, mice were i.n. administered 500 VFU of PR8-OTII influenza virus. In some experiments, mice received i.p. administration of 200 μL of 2.5 mg/mL EDU (Invitrogen) or 200 μg of AZD9898 (MedChem Express). For intra- and perivascular staining, mice were i.v. administered 3 μg anti-CD45.2 antibody (clone 104, BD Biosciences) 5 min before euthanasia and tissue harvest.

### Flow cytometry and cell sorting

Lungs were isolated, cut into small fragments, and digested for 45 min at 37 °C with 0.6 mg/mL collagenase A (Sigma) and 30 mg/mL DNase I (Sigma) in RPMI-1640 medium (GIBCO). Digested lungs or mLNs were mechanically disrupted by passage through a wire mesh. Blood was collected in Dextran-EDTA buffer. Bone marrow was collected from the femurs and tibias and filtered through a wire mesh. Red blood cells were lysed with 150 mM $NH_4Cl$, 10 mM $KHCO_3$, and 0.1 mM EDTA. Fc receptors were blocked using anti-mouse CD16/32 (clone 93; BioXCell, BE0307), followed by staining with fluorochrome-conjugated antibodies against surface and intracellular markers. Antibodies included CD4 (GK1.5; BD Biosciences, 553729), CD11b (M1/70; BD Biosciences, 553311), Ly6G (1A8; BioLegend, 127603), MHC-II (M5/114.15.2; BioLegend, 107620), and Siglec-F (E50-2440; BD Biosciences, 565526), among others. Unless otherwise noted, antibodies were used at manufacturer-recommended dilutions (typically 1:100 for surface staining and 1:50–1:100 for intracellular staining). A full list of all antibodies, clones, catalog numbers, RRIDs, and dilutions is provided in the Reporting Summary. Dead cells were excluded using 7AAD (Sigma-Aldrich). For intracellular cytokine staining, cell suspensions were stimulated with PMA (20 ng/mL) and calcimycin (1 mg/mL) in the presence of BD GolgiPlug for 5 h, followed by surface staining, fixation, and permeabilization using the BD Cytofix/Cytoperm Plus Kit. Intranuclear staining was performed using the Mouse Regulatory T Cell Staining Kit (eBioscience). EdU incorporation was measured with the Click-iT Plus EdU Flow Cytometry Assay Kit (Invitrogen). Data were acquired on an Attune NxT or BD LSRFortessa and analyzed with FlowJo v10.10. Cell sorting was performed on a BD FACSAria II (post-sort purity >98%). In all flow cytometry dot plots, scale bars include a darker reference bar indicating the 0 baseline, with signal extending up to $10^6$.

### OTII cell transfer

CD4+ T cells were isolated from the spleens of naïve 4get.OTII TCR-transgenic mice using MACS (Miltenyi Biotec) or the EasySep CD4+ Positive Selection Kit (STEMCELL Technologies) (post-isolation purity >95%). Equivalent numbers (~50,000) of naïve OTII cells were transferred i.v. into naïve congenic recipients one day before OVA treatments.

### Cell morphology

100 μL of sorted cells (1×$10^5$ cells/mL) were loaded into cuvettes attached to micro slides and spun at 500–600 rpm for 5 min. The cuvette was then removed, and the slides were stained using the Hema 3 Stat Pack (Fisher Scientific). Slides were covered and sealed with

Cytoseal Mounting Medium (Electron Microscopy Sciences). Images were captured using an Echo Revolve 4 microscope.

## Immunofluorescence microscopy

Lungs from Nur77-GFP mice were fixed overnight in 1% paraformaldehyde (Electron Microscopy Sciences) at 4 °C, washed with PBS and ammonium chloride, and cryoprotected in 30% sucrose overnight. Tissues were embedded in OCT (Fisher Healthcare), and 12 μm cryosections were prepared. Sections were blocked with 5% donkey serum (Sigma-Aldrich) and stained with antibodies against CCR7 (4B12; eBioscience, 13-1971-82), CD31 (MEC13.3; BioLegend, 102516), CX3CR1 (SA011F11; BioLegend, 149005), Ly6G (1A8; BioLegend, 127603), LYVE-1 (ALY7; eBioscience, 53-0443-82), GFP (A21311; Life Technologies), and DAPI. Slides were mounted with ProLong Diamond Antifade Mountant (Thermo Fisher Scientific). Images were captured on a Leica SP8 confocal microscope and analyzed with LAS X software (v3.5.21594.6).

## Migration assays

100 ng of CCL21 or CCL19 in RPMI medium with 1% FBS was added to the bottom chamber of each 24-well Transwell plate (polycarbonate filter with a 5-μm pore; Costar). mDCs ($50 \times 10^4$ cells per Transwell) were added to the upper chamber in the presence or absence of 100 nM LTC4 or LTD4, followed by incubation for 90 min at 37 °C. Cells that had transmigrated were collected from the lower chamber and counted. Results are presented as the number of input cells that migrated in the assay. No spontaneous migration in response to the control medium was observed.

## Cell culture and stimulation

Sorted Ly6G+ macrophages were cultured in 96-well flat-bottom plates pre-coated with fibronectin (10 μg/mL) in the presence of M-CSF (10 ng/mL), IL-3 (5 ng/mL), TGF-β (1 ng/mL), and IL-10 (10 ng/mL) and incubated overnight at 37 °C. The following day, cells were stimulated for 1 h at 37 °C with control media or papain (20–100 μg/mL). Plates were then placed on ice, and supernatants were collected for LTC4 quantification by ELISA.

## ELISA

LTC4 was quantified in lung homogenates and cell culture supernatants using the Leukotriene C4 ELISA Kit (Cayman Chemical Company), following the kit instructions.

## RNA sequencing (RNA-seq)

**Primary analysis.** Library preparation and RNA-seq were conducted through Genewiz (Azenta Life Sciences). Libraries were sequenced on the HiSeq 3000 platform. The quality of raw sequence fastq-formatted files was assessed using fastQC (https://www.bioinformatics.babraham.ac.uk/projects/fastqc/). Sequences were trimmed using Trim Galore (version 0.4.4) with phred33 scores, paired-end reads, and the Nextera adapter options (https://github.com/FelixKrueger/TrimGalore). Trimmed sequences were aligned using STAR (version 2.5.2a) with mouse GRCm39 and default options[49]. Aligned reads were counted with HTseq-count (version 0.6.1p1) set for unstranded reads using the GRCm39 annotation file[50].

**Downstream analysis**
The R package edgeR version 4.4.1[51,52] was used to assess pairwise differential expression between sample groups. Only genes with counts per million greater than 1 for at least three samples were considered for further analysis. Comparison of our data to other published datasets was accomplished using gene set enrichment analysis (GSEA, version 4.3.2)[53]. Pathway and function enrichment analyses were performed using Ingenuity Pathway Analysis (IPA, Qiagen Inc. (https://digitalinsights.qiagen.com/IPA)[54]. Differentially expressed genes (DEGs) were identified based on the criteria of FDR < 0.001 and log2 fold-change > 1 or < −1.

## Single cell RNA sequencing (scRNA-seq)

Cell suspension and 10x barcoded gel beads were loaded into 10x Chromium™ Single Cell Chip G (PN 1000120) to capture single cells in Nano liter-scale oil droplets by 10xGenomics Chromium X Controller using Chromium Next GEM Single Cell 3′ v3.1 reagents (PN-1000268). The single-cell 3′ v3 gene expression library was sequenced on a NovaSeq 6000 101 cycle + 101 cycle symmetric run. The OpenOmics/cell-seek pipeline v2.0.1 and v3.0.2 (https://doi.org/10.5281/zenodo.11106856), leveraging 10x Genomics Cell Ranger v8.0.0 and v9.0.0 generated matrices from FASTQ files using the 10x Genomics GRCm39-2024-A transcriptome reference and performed Seurat QC filtering using Seurat v5.1.0 and v5.2.1[55] in R v4.3.3 and v4.4.0 (https://www.R-project.org/). The Seurat filtering was performed to remove low-count cells and potential doublets by applying 3 median absolute deviations from the median log-library size filter to the gene and read counts. The same filtering was applied to the mitochondrial percentage with only an upper threshold. Cells were log10 normalized, dimension reduction performed using 30 principal components, and data visualized using Uniform Manifold Approximation and Projections (UMAP). For the allergen+ dataset, unsupervised clustering was performed at resolution 0.6, and monocyte/macrophage populations were isolated by subsetting clusters 3, 4, 5, 7, and 11. The subset was re-normalized, scaled, and re-clustered at resolution 0.4. For the comparative analysis of Ly6G+ MΦ and ncMo, datasets were first merged and clustered at a resolution of 0.2. Based on marker inspection, clusters 0–2 and 5–7 were selected and reanalyzed. Samples were maintained as separate layers and reprocessed with 30 PCs using Seurat's joint PCA integration workflow. After integration, reclustering was performed at resolution 0.2, and cluster 2 was further subclustered at resolution 0.1. Top marker analysis of the Seurat clusters and branches was performed using Model-based Analysis of Single-cell Transcriptomics (MAST) (https://github.com/RGLab/MAST/) and the Bonferroni test for multiple test correction. Top markers with an average log2 fold-change of greater than 0.5 and adjusted p-value less than 0.05 were used as input for Over-Representation Analysis (ORA). ORA was performed using the R package clusterProfiler v4.10.1 and v4.14.6[56] with the GO Biological Process and KEGG databases. Gene Set Enrichment Analysis was performed using the Bioconductor package UCell v2.6.2[57]. Trajectory analysis was performed using Monocle 3 v1.3.7[58] with closed loops set to false. Feature plots, UMAPs, and violin plots were visualized using the R package scCustomize v3.0.1 (https://zenodo.org/records/14529706). The dot plot and heatmap were visualized using the Bioconductor package ComplexHeatmap v2.18.0[59]. Cell cycle scoring was performed with Seurat using mouse G2/M and S phase markers.

## Statistical analysis

Data analysis was performed using GraphPad Prism (Version 10.2.0). Statistical tests used are indicated in the figure legends. P-values of less than 0.05 were considered statistically significant (*$P < 0.05$, **$P < 0.01$, ***$P < 0.001$, **$P < 0.0001$).

## Reporting summary

Further information on research design is available in the Nature Portfolio Reporting Summary linked to this article.

## Data availability

RNA-seq data have been deposited in the Gene Expression Omnibus (GEO) under accession GSE291081, and scRNA-seq data have been deposited in GEO under accessions GSE289548 and GSE305119 and are publicly available. Any additional information required to reanalyze

the data reported in this paper is available from the corresponding author upon request. All other data are available in the article and its Supplementary files or from the corresponding author upon request. Source data are provided with this paper.

## Code availability

The code used to analyze the scRNA-seq data can be accessed at https://doi.org/10.5281/zenodo.15149503 and https://doi.org/10.5281/zenodo.16848446.

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

## Acknowledgements

We thank Dr. Catherine C. Hedrick (Augusta University, Augusta, GA, USA), Pamela A. Frischmeyer-Guerrerio, Karen Laky, and Justin Lack (National Institute of Allergy and Infectious Diseases, National Institutes of Health, Bethesda, MD, USA) for providing mice and bioinformatics support. We also thank Becca Burnham, Kelsey Browning, and Thomas 'Scott' Simpler (University of Alabama at Birmingham, Birmingham, AL, USA) for animal husbandry, and the UAB FCSC Core for assistance with cell sorting and preparation of scRNA-seq libraries. This work utilized the computational resources of the NIH HPC Biowulf cluster (https://hpc.nih.gov). This research was supported in part by National Institutes of Health (NIH) grant 2R01AI116584 to BL, and by the Intramural Research Program of the NIH. The contributions of the NIH author(s) are considered Works of the United States Government. The findings and conclusions presented in this paper are those of the author(s) and do not necessarily reflect the views of the NIH or the U.S. Department of Health and Human Services.

## Author contributions

Conceptualization, A.M. and B.L.; Methodology, A.M., H.B., C.L., S.D., G.P., J.C.G., and B.L.; Formal analysis, A.M., B.D., D.D.H., P.L., A.F.R., and B.L.; Visualization, A.M., B.D., and B.L.; Writing – original draft, A.M. and B.L.; Writing – review and editing, B.L.; Supervision, B.L.; Funding acquisition, B.L.

## Funding

## Competing interests

The authors declare no competing interests.
