## [Transparent Peer Review file · Nature Communications]

Atypical pericapillary Ly6G⁺Nur77⁺ macrophages initiate type-2 immune responses to allergens in the lung

Corresponding Author: Dr Beatriz León

Version 0:

Reviewer comments:

Reviewer #1

(Remarks to the Author)

The manuscript by Meloun et al. presents a compelling study uncovering a novel function of Ly6G⁺ perivascular macrophages in the initiation of type 2 immune responses in the lung. By elucidating a PAR2–NR4A1–ALOX5–LTC₄ signaling cascade that promotes dendritic cell (DC) migration and subsequent Th2 cell differentiation, this work provides mechanistic insights into how innate cells sense and respond to allergen exposure. The study is methodologically sound, conceptually novel, and highly relevant to our understanding of type 2 immunity and allergic inflammation. That said, there are a few aspects where the authors could further strengthen their conclusions and enhance the impact of this already promising work.

Major Comments

1. Clarification of the direct role of PAR2 activation in NR4A1 induction

The manuscript proposes that HDM-derived cysteine proteases cleave PAR2 in Ly6G⁺ macrophages, leading to NR4A1 induction and subsequent ALOX5 expression and LTC₄ production. While this signaling axis is supported by in vivo genetic loss-of-function experiments, the mechanistic link between PAR2 cleavage and NR4A1 activation remains indirect. To strengthen this critical aspect of the model, the authors are encouraged to:

- Provide direct evidence of PAR2 cleavage, for example using cleavage-specific antibodies in Western blot analysis.
- Examine whether ALOX5 expression and LTC₄ production are induced following in vitro stimulation of sorted Ly6G⁺ macrophages with HDM or papain.
- Determine whether these responses are abrogated in PAR2-deficient cells or upon treatment with protease inhibitors.

These additional experiments would help establish a more direct causal relationship between PAR2 activation and the downstream transcriptional and metabolic responses.

2. Elucidation of whether Ly6G⁺ macrophages and antigen-loaded DCs physically interact

It remains unclear whether the effect of Ly6G⁺ macrophages on DCs is mediated via direct contact or paracrine signaling. Including imaging data to assess spatial proximity or contact between Ly6G⁺ cells and OVA⁺ DCs would help clarify the nature of this cellular interaction.

3. Anatomical localization of Ly6G⁺ macrophages near lymphatic vessels

Given that tissue DCs typically migrate to draining lymph nodes via afferent lymphatic vessels, it would be valuable to demonstrate whether Ly6G⁺ macrophages are anatomically positioned near lymphatic structures (e.g., LYVE1⁺ or Prox1⁺ vessels), particularly at sites where DCs initiate egress. Triple immunostaining for Ly6G, ALOX5, and lymphatic markers would support this point.

4. Retention of antigen-loaded DCs in the lung upon LTC₄ or ALOX5 inhibition

If Ly6G⁺ macrophages promote DC migration via LTC₄, inhibition of this pathway should result in increased retention of antigen-loaded DCs in the lung. While the authors show an accumulation of total CD11c⁺ cells, it is not clear whether these include antigen-bearing DCs. Tracking OVA⁺ DCs (e.g., via OVA-AF647 labeling) under ALOX5-deficient or LTC₄-inhibited conditions would directly support the proposed mechanism.

5. Th2 cell transfer to distinguish initiation vs. effector stages

To confirm that Ly6G⁺ macrophages are required specifically for initiating Th2 differentiation (rather than affecting effector

cell function), the authors could assess whether adoptively transferred pre-polarized OT-II Th2 cells bypass the requirement for Ly6G⁺ macrophages or LTC₄. This would pinpoint the step at which this innate mechanism operates.

Minor Comments

1. Characterization of Ly6G⁺ macrophages

The Ly6G⁺ macrophages shown in Extended Data Fig. 2c appear to display a binucleated morphology. Similar nuclear features have been observed in certain atypical monocyte populations, such as the SatM cells described in a fibrosis model (Sato et al., *Nature*, 2017; PMID: 28002407). While the relevance may be limited, briefly acknowledging this morphological similarity might help place these cells within the broader diversity of monocyte/macrophage subsets.

2. The selectivity of PAR2 over other PARs

While the authors focus on PAR2, it is worth noting that HDM-derived cysteine proteases can potentially activate multiple members of the PAR family (PAR1–4). Notably, recent studies have shown that PAR1 is highly expressed in Th2 cells and contributes to allergic inflammation by enhancing IL-5 and IL-13 production (Kumagai et al., *PNAS*, 2023; PMID: 38015852). Although additional experiments are not essential, a brief discussion on the potential roles or expression levels of other PARs—particularly PAR1—could help contextualize the specificity of the PAR2-dependent mechanism highlighted in this study.

Reviewer #2

(Remarks to the Author)

The study by Meloun et al. identifies an as-yet-unknown sensor of allergens with protease activity and an important actor of type 2 immunity to allergens, namely atypical macrophages expressing PAR2 and Ly6G. The experimental design is robust and appropriate, the experiments are very well executed, the amount of data is impressive and they support most of the authors' conclusions. The manuscript is very well written and easy to read, and the authors should be commended for their high-quality work. Since such Ly6G⁺ macrophages have not been previously described in the healthy mouse lung, I think that their characterization at steady-state warrants some additional investigation, especially regarding their localization. Comments from a to e below are related to this.

Major comments:

- a. My understanding of the data in Fig. 3 is that Ly6G⁺ Macs at steady-state derive from GMPs postnatally and are not rapidly replaced by circulating monocytes. The authors use the word “tissue-resident” – what is their definition of tissue-residence? This should be clarified, as the possibility remains that they are intravascular (see below).
- b. Related to this, it seems that Ly6G⁺ macs are relatively long-lived. EdU-pulse experiments could provide insights into their half-life.
- c. If they derived postnatally from GMPs, as shown with the Ms3a3Tdtom mice, when do they appear in the lung after birth?
- d. In Fig. 3y, an important missing control is the chimeric data in non papain-injected mice. The authors state that accumulation of Ly6G⁺ macs in the lung after treatment with protease allergens requires PAR2, but they gate on papain+ cells, not all the cells, and they only look at treated mice, not naïve mice.
- e. My main conceptual concern relates to the localization of these cells. The authors state that Ly6G⁺ Macs are “perivascular” based on anti-CD45 iv staining, then later they say that Nr4a1 + Ly6G⁺ Macs are primarily present in the alveolar interstitium, located between the alveolar epithelium and the CD31+ capillary endothelium. First, to me, “perivascular” would rather refer to their preferential localization around bigger blood vessels (arterioles, venules), not capillaries, while the alveolar interstitium is not “perivascular” per se. I think that this is particularly important to clarify since it also relates to the site of sensitization to protease allergens and DC uptake? Does it occur in the alveoli? Or rather at the level of bronchial epithelium, which is suggested in many review and original articles in the field? Second, I understand that the confocal microscopy pictures shown in Extended Data Fig. 4c are taken from a naïve mouse lung. If correct, Nr4a1+Ly6G⁺ “Macs” are strikingly abundant on the slide, while they are a very rare population. An alternative, more likely, scenario is that the Nr4a1+Ly6G⁺ lung cells are lung neutrophils, which also express Nr4a1 (unpublished observations – and this does not fit with the data shown in Extended Data Fig. 4b – are Neu from the lung, as stated? or from the blood?). Third, it is impossible to draw the conclusion, based on the pictures shown, that Ly6G⁺ Macs are not intravascular, especially since they stain positive for anti-CD45 iv. Hence, it is imperative to re-assess the localization of Ly6G⁺ Macs at steady-state using additional markers: the best would be the use of Cx3cr1 GFP mice, or alternatively, the use of CD68, with a particular focus at peribronchial/perivascular locations.
- f. All the “data not shown” should be included in the manuscript, even though I appreciate already the substantial amount of data.

Comments on figures:

- Figure 2a–f: The authors aim to evaluate the role of HDM proteases in promoting mDC migration. Why was heat-inactivated HDM used instead of HDM pretreated with E-64, a selective inhibitor of cysteine protease activity? This alternative would more precisely address the role of cysteine proteases.
- Extended Data Figure 2b: The gating on histograms appears to be suboptimal. Given the high autofluorescence of macrophages, it is recommended to use Fluorescence Minus One (FMO) controls to accurately define positive populations. Similarly, for EdU staining, cells from non-injected mice should be subjected to the Click reaction to serve as a proper negative control.
- Figure 4i: Gating on density plots is preferable, as it facilitates clearer discrimination between positive and negative populations. Furthermore, cells from PBS-injected control mice should be included as a negative control for CD45

intravenous staining.

- Extended Data Figure 3a-b: the moDC transcriptome fits perfectly with the signature of lung interstitial macrophages (IM), as they express all the macrophages and IM-related markers (*Mafb*, *Fcgr1*, *C1q*, *Mrc1*, etc), but do not express DC-specific markers. Even though the discrimination between moDCs and IMs remain contentious, I am curious why the authors chose to name these cells moDCs.
- Extended Data Figure 3e: Instead of gating on CD11b⁺⁺ cells, staining for Ly6G would provide a more accurate identification of the target population.
- Figure 4b: The authors demonstrate that *Nr4a1* is necessary for the accumulation of Ly6G⁺ macrophages and ncMo. However, it should be considered that ncMo may drive the differentiation of Ly6G⁺ macrophages upon allergen exposure. Therefore, the observed reduction in Ly6G⁺ macrophages in these mice may be secondary to the absence of ncMo, rather than reflecting a direct requirement of *Nr4a1* in Ly6G⁺ macrophages themselves. This should be clarified.
- Figures 4c–d: The authors conduct mixed bone marrow chimeras (WT/*se_2*^{-/-}) to assess allergen uptake by Ly6G⁺ macrophages in the absence of *Nr4a1*, showing reduced papain capture by Ly6G⁺ macrophages from *se_2*^{-/-} origin. To strengthen these findings, it would be important to report the chimerism levels in Ly6G⁺ macrophage population. If Ly6G⁺ macrophages require *Nr4a1* for differentiation, then cells of *se_2*^{-/-} origin should fail to differentiate into Ly6G⁺ macrophages, and this underrepresentation may account for the reduced frequency of papain⁺ Ly6G⁺ macrophages observed in Figure 4c.

Further suggestions for improvement can be found below.

- Across all experiments, the figure panels are very small and should be enlarged with a focus on the most important findings (some gating strategies could be moved to supplement) for better readability.
- All the data converge to the possibility that ncMo give rise to Ly6G⁺ Macs. This is particularly interesting since a previous study identified CD16.2⁺ ncMo in the mouse lung, also expressing *Ace* and *Plac8*, like Ly6G⁺ Macs. This possibility could be discussed.
- Recent findings have shown that (likely other) recruited Ly6G⁺ macrophages can play a role in alveolar repair following influenza virus infection, and such Ly6G⁺ macs depend on both GM-CSF and type 2 cytokine signaling. Since cysteine proteases have previously been reported to induce GM-CSF production in non-classical monocytes (ncMo) (Kaur et al., Cell Reports, 2021), and since alternative activation via IL4/13 pops up in the transcriptomic data, it would be a nice mechanistic addition to investigate the extent to which Ly6G⁺ macrophages rely on these signaling pathways. Alternatively, this possibility could be discussed.

Reviewer #3

(Remarks to the Author)

Meloun and colleagues report, for the first time, that in an allergic model of pulmonary inflammation, Ly6G⁺ Macrophages produce CysLTs via PAR2, which facilitates dendritic cell migration to mediastinal lymph nodes (mLN). This study presents several novel findings, including the presence of Ly6g⁺ Macrophages, similar to those reported in an IAV model of lung inflammation (REF 30). The activation of Ly6G⁺ macrophages through *Nr4a1* and PAR to produce CysLTs. And the effects of CysLTs on DCs migration to mLN to activate T-cells. The methodology includes the use of various chimeras and single-cell sequencing. The data presented support the main conclusions. I offered several observations and recommendations to further refine and expand the study.

The authors note in the discussion that lung Ly6g⁺ macrophages have previously been identified (Ref. 30). There are reported similarities between the Ly6G⁺ macrophages described in this study and those in Ref. 30 identified in a viral model of lung inflammation, including their origin from GMP precursors and their alveolar location. The authors may wish to consider investigating additional potential similarities, such as whether papain-induced Ly6g⁺ macrophages express *Arg1*. Are they short-lived, meaning are they only involved in sensitization? Additionally, how CysLTs enhance DC migration in response to chemokines?

Figure 3w: The authors state that “papain⁺ Ly6G⁺ MΦ did not accumulate in the lungs of treated *Par2*^{-/-} mice compared to WT”; however, there appears to be only a 50% reduction. Furthermore, given that moDCs were also reduced in *PAR2*^{-/-}, is PAR2 necessary for cMo or non-classical Mo differentiation into Ly6G⁺ macrophages, or for their proliferation?

Extended Data 3b. Although Ly6G⁺ macrophages and non-classical monocytes share certain genes, non-classical monocytes were not identified in the transcriptional signature clusters.

Line 364. It would be informative to review the data on HDM, even if just as Extended Data.

In the BM chimera with WT and *Alox5*^{-/-} BM donors, papain sensitization yields minimal lung LTC4 (~5 ng/ml). Yet, in figure 6n (WT/A-) under the same conditions, LTC4 levels reach about 40 ng/ml. What accounts for this difference?

Figure 6k. Ly6g⁺ macrophage counts are comparable in s/A^{-/-} chimeras and controls. However, does ALOX5 deficiency alter their activation, proliferation, or recruitment?

Line 455. The statement that the three CysLT receptors “have redundant functions” is inaccurate. CysLT1, CysLT2, and CysLT3 each preferentially bind LTD4, LTC4, and LTE4, respectively, resulting in distinct effects based on cell type and receptor expression.

Figure 7. Asthmatic patients are already sensitized to allergens. Does AZD9898 given during the challenge phase suppress T-cell responses, and are these results relevant to real allergens like house dust mites (HDM)?

Does AZD9898 affect Ly6G⁺ macrophage numbers or activation?

Line 513: The authors discuss Ruscitti et al. (ref 30) and indicate differences but do not highlight similarities in Ly6G⁺ macrophages, especially in relation to their location and development. Including this information would provide readers with a more comprehensive understanding.

Minor

Line 349: add "Fig. 5b"

Line 374: add "Fig. 5h"

Version 1:

Reviewer comments:

Reviewer #2

(Remarks to the Author)

The authors have performed a tremendous amount of work during these revisions. In my opinion, the manuscript is now fully acceptable for publication in Nature Communications, and the authors should be commended for their impressive work.

Reviewer #3

(Remarks to the Author)

I would like to express my gratitude to the Authors for their thorough response to my inquiries and for expanding their dataset. I am particularly pleased to note the identification of Ly6G⁺ macrophages as long-lived cells and their developmental relationship with non-classical monocytes (ncMo).

I agree with Review 2 and the authors' decision to clarify the description of "perivascular" macrophages, specifying their localization to the alveolar-pericapillary region. This description should be consistently incorporated throughout the manuscript, including the title, to enhance clarity and precision.

Figure 6. The finding that Ly6G⁺ macrophages produce LTC₄, which subsequently induces CCR7⁺ dendritic cells (DC) to migrate toward CCL21, represents an intriguing mechanism. Although this may extend beyond the scope of the current study, it would be interesting to investigate which receptor is specifically involved in the novel Ly6G⁺ macrophage-induced activation of DC.

Regarding Figure 3v, I appreciate the authors' response to my comment. However, I maintain that the phrase "did not accumulate" might be an overstatement, given the comparable reduction observed in DC.

Regarding Figure 6k, while it may extend beyond the primary scope of this study, considering the potential autocrine effects of CysLTs, I am curious whether the absence of Alox5 influences the production of other molecules by these macrophages upon activation.

We sincerely thank the Editor and Reviewers for their thoughtful and constructive comments, which have greatly improved the quality and clarity of our manuscript. In this revised version, we have carefully addressed all the points raised. Specifically, we have experimentally assessed the direct function of PAR2, clarified the longevity and steady-state presence of Ly6G⁺ macrophages, and incorporated new data and discussion regarding their localization. We have also performed AZD8989 inhibitor experiments in sensitized animals, expanded the discussion of potential developmental pathways, and comprehensively revised the text to address all other reviewer comments with new data, updated figures, and clarifying explanations throughout the manuscript.

REVIEWER COMMENTS

Reviewer #1 (Remarks to the Author):

The manuscript by Meloun et al. presents a compelling study uncovering a novel function of Ly6G⁺ perivascular macrophages in the initiation of type 2 immune responses in the lung. By elucidating a PAR2–NR4A1–ALOX5–LTC₄ signaling cascade that promotes dendritic cell (DC) migration and subsequent Th2 cell differentiation, this work provides mechanistic insights into how innate cells sense and respond to allergen exposure. The study is methodologically sound, conceptually novel, and highly relevant to our understanding of type 2 immunity and allergic inflammation.

That said, there are a few aspects where the authors could further strengthen their conclusions and enhance the impact of this already promising work.

Major Comments

1. Clarification of the direct role of PAR2 activation in NR4A1 induction

The manuscript proposes that HDM-derived cysteine proteases cleave PAR2 in Ly6G⁺ macrophages, leading to NR4A1 induction and subsequent ALOX5 expression and LTC₄ production. While this signaling axis is supported by *in vivo* genetic loss-of-function experiments, the mechanistic link between PAR2 cleavage and NR4A1 activation remains indirect. To strengthen this critical aspect of the model, the authors are encouraged to:

- Provide direct evidence of PAR2 cleavage, for example using cleavage-specific antibodies in Western blot analysis.
- Examine whether ALOX5 expression and LTC₄ production are induced following *in vitro* stimulation of sorted Ly6G⁺ macrophages with HDM or papain.
- Determine whether these responses are abrogated in PAR2-deficient cells or upon treatment with protease inhibitors.

These additional experiments would help establish a more direct causal relationship between PAR2 activation and the downstream transcriptional and metabolic responses.

We agree that establishing a more direct connection between PAR2 activation by protease allergens and downstream LTC₄ synthesis in Ly6G⁺ macrophages would strengthen our model. We have now added new *in vitro* data using sorted Ly6G⁺ macrophages, as well as supporting *in vivo* observations:

- **Papain induces LTC₄ synthesis in Ly6G⁺ macrophages in a PAR2-dependent manner.** Sorted Ly6G⁺ macrophages were stimulated with papain (20-100 µg mL⁻¹) for

1 h. LTC₄ levels in the supernatant increased in WT but not in Par2^{-/-} cells (**new Extended Data Fig. 7d**; see lines 446–450). This experiment directly addresses the reviewer's request to test *in vitro* LTC₄ induction by protease allergens and to demonstrate its deficiency in PAR2-deficient Ly6G⁺ macrophages. The rapid kinetics are consistent with *de novo* eicosanoid synthesis, supported by the constitutive expression of Alox5 and Ltc4s in Ly6G⁺ macrophages (**Extended Data Fig. 7a-c**). Because sorted Ly6G⁺ macrophages did not survive extended culture, we could not reliably assess whether PAR2 activation further increased Ltc4s transcription, as observed *in vivo* (**Fig. 6d**), nor could we evaluate other long-term effects of PAR2 activation *in vitro*.

- **PAR2 surface expression is down-modulated following protease stimulation.** By flow cytometry, surface PAR2 expression decreased on WT Ly6G⁺ macrophages *in vivo* 24 h after HDM or papain exposure compared with naïve mice (**new Extended Data Fig. 3b**; see lines 222–225), suggesting activation-associated receptor internalization or down-modulation. We fully agree that cleavage-specific assays (e.g., neo-epitope antibodies or Western blot) would more precisely define the biochemical step. However, such assays were not feasible with our limited primary cell yields within this revision.

Together with the *in vivo* genetic loss-of-function data, these findings provide direct functional evidence linking protease-dependent PAR2 activation to early eicosanoid output in Ly6G⁺ macrophages. As a note, Ly6G⁺ macrophages constitutively express NR4A1, PAR2, ALOX5, and LTC₄ S, suggesting that they possess a transcriptionally poised machinery for leukotriene synthesis. However, LTC₄ production is only triggered upon protease-dependent PAR2 activation and requires NR4A1 expression. Both NR4A1 and PAR2 are therefore essential for this response, although their mechanistic relationship remains to be defined. We have clarified this point in the revised discussion (see lines 555–557).

2. Elucidation of whether Ly6G⁺ macrophages and antigen-loaded DCs physically interact
It remains unclear whether the effect of Ly6G⁺ macrophages on DCs is mediated via direct contact or paracrine signaling. Including imaging data to assess spatial proximity or contact between Ly6G⁺ cells and OVA⁺ DCs would help clarify the nature of this cellular interaction.

3. Anatomical localization of Ly6G⁺ macrophages near lymphatic vessels
Given that tissue DCs typically migrate to draining lymph nodes via afferent lymphatic vessels, it would be valuable to demonstrate whether Ly6G⁺ macrophages are anatomically positioned near lymphatic structures (e.g., LYVE1⁺ or Prox1⁺ vessels), particularly at sites where DCs initiate egress. Triple immunostaining for Ly6G, ALOX5, and lymphatic markers would support this point.

Points 2 & 3: Anatomical localization of Ly6G⁺ macrophages and DCs in the lung.

We thank the reviewer for these insightful comments regarding the spatial relationship between Ly6G⁺ macrophages and dendritic cells (DCs) in the lung. To investigate whether these cell types are spatially associated, we performed additional immunofluorescence analyses of lung sections from naïve and papain-treated mice. In naïve lungs, CCR7⁺ DCs were sparsely distributed throughout the parenchyma and frequently located in close proximity to Nr4a1⁺ Ly6G⁺ macrophages near CD31⁺ capillaries (**new Extended Data Fig. 6h,i**). Following papain treatment, however, CCR7⁺ DCs formed prominent clusters around LYVE1⁺ lymphatic vessels (**Extended Data Fig. 6h**). Thus, Ly6G⁺ macrophages are primarily associated with the CD31⁺ capillary network within the alveolar-airway parenchyma, where DCs can also localize, particularly in naïve lungs, likely allowing them to access inhaled antigens. Together with our

functional data showing that Ly6G⁺ macrophage activation upon allergen exposure promotes DC migration to the mediastinal lymph nodes through the production of cysteinyl leukotrienes (cysLTs), these histological observations suggest that activated Ly6G⁺ macrophages communicate locally with DCs in the alveolar-airway parenchyma to enhance their CCR7-dependent chemotactic migration toward lymphatic vessels. This model is consistent with our *in vivo* migration assays showing that cysLT signaling enhances CCR7-dependent DC chemotaxis toward CCL21 (**Fig. 6p,q**). Together, these findings support a model in which activated Ly6G⁺ macrophages located in the perivascular-interstitial space locally instruct DCs through the production of cysteinyl leukotrienes, thereby enhancing their CCR7-dependent responsiveness to CCL21 and guiding their migration toward CCL21-producing LYVE1⁺ lymphatic vessels for egress to the draining lymph nodes (**see lines 419-425**).

4. Retention of antigen-loaded DCs in the lung upon LTC₄ or ALOX5 inhibition

If Ly6G⁺ macrophages promote DC migration via LTC₄, inhibition of this pathway should result in increased retention of antigen-loaded DCs in the lung. While the authors show an accumulation of total CD11c⁺ cells, it is not clear whether these include antigen-bearing DCs. Tracking OVA⁺ DCs (e.g., via OVA-AF647 labeling) under ALOX5-deficient or LTC₄-inhibited conditions would directly support the proposed mechanism.

We tracked papain⁺ DCs in the lung and mediastinal lymph node of Alox5^{-/-} mice, Alox5^{-/-} mixed bone marrow chimeras, and after pharmacological inhibition of LTC₄ synthase with AZD9898. Although we observed a marked reduction in the migration of allergen-bearing cells to the draining lymph node, this was not accompanied by a detectable increase in the proportion or relative accumulation of these cells in the lung (**Fig. 6h; Extended Data Fig. 7j,k**). This likely reflects the fact that the total resident DC pool in the lung is much larger than the small CCR7⁺ subset that migrates to lymph nodes; thus, even when cysLT-dependent egress is impaired, the relative change in total or antigen-bearing DCs within the lung remains below detection. In addition, DCs with delayed migration may undergo apoptosis or be cleared rather than accumulating as intact antigen⁺ cells. Therefore, the most informative measure of cysLT-dependent DC trafficking is the arrival of allergen-bearing DCs in the draining lymph node, rather than their relative retention in the lung.

5. Th2 cell transfer to distinguish initiation vs. effector stages

To confirm that Ly6G⁺ macrophages are required specifically for initiating Th2 differentiation (rather than affecting effector cell function), the authors could assess whether adoptively transferred pre-polarized OT-II Th2 cells bypass the requirement for Ly6G⁺ macrophages or LTC₄. This would pinpoint the step at which this innate mechanism operates.

We appreciate this thoughtful suggestion. Our data indicate that Ly6G⁺ macrophages act upstream during the initiation phase of the type-2 response by promoting CCR7-dependent dendritic cell migration required for efficient naïve CD4⁺ T-cell expansion. When Ly6G⁺ macrophage activation or the LTC₄ pathway is disrupted, naïve antigen-specific T cells fail to expand efficiently due to defective DC migration; however, the small fraction of T cells that do receive adequate stimulation still polarize to Th2, as IL-4-producing cells are preserved among the residual responders. Thus, the dominant defect lies in T-cell priming and expansion, not in Th2 effector differentiation.

While adoptive transfer of pre-polarized OT-II Th2 cells could, in principle, bypass the initiation step, interpretation of such experiments would be complicated by the fact that *in vitro*-polarized Th2 cells exhibit altered and non-physiological homing patterns. Our data already demonstrate

impaired expansion with intact per-cell Th2 polarization within the draining lymph node. We therefore focused on this aspect, recognizing that further assessment of effector Th2 homing to the lung or their local effector functions would involve distinct mechanisms beyond the scope of the present study.

Minor Comments

1. Characterization of Ly6G⁺ macrophages

The Ly6G⁺ macrophages shown in Extended Data Fig. 2c appear to display a binucleated morphology. Similar nuclear features have been observed in certain atypical monocyte populations, such as the SatM cells described in a fibrosis model (Sato et al., Nature, 2017; PMID: 28002407). While the relevance may be limited, briefly acknowledging this morphological similarity might help place these cells within the broader diversity of monocyte/macrophage subsets.

Thank you for this insightful comment. We agree that it is important to acknowledge this study and have now incorporated this point into the Discussion (paragraph, lines 564–571, with the reference added in line 566). This will be an interesting direction for future studies to further explore potential relationships among these monocyte/macrophage subsets.

2. The selectivity of PAR2 over other PARs

While the authors focus on PAR2, it is worth noting that HDM-derived cysteine proteases can potentially activate multiple members of the PAR family (PAR1–4). Notably, recent studies have shown that PAR1 is highly expressed in Th2 cells and contributes to allergic inflammation by enhancing IL-5 and IL-13 production (Kumagai et al., PNAS, 2023; PMID: 38015852). Although additional experiments are not essential, a brief discussion on the potential roles or expression levels of other PARs—particularly PAR1—could help contextualize the specificity of the PAR2-dependent mechanism highlighted in this study.

We appreciate this insightful suggestion. Although cysteine proteases such as Der p 1 may have the potential to cleave multiple PARs under certain experimental conditions, our study specifically demonstrates that activation of Ly6G⁺ macrophages and the subsequent initiation of Th2 responses are mediated through PAR2. Given that our data clearly establish a dominant and specific requirement for PAR2, we believe that expanding the discussion to include other PARs would not enhance the mechanistic clarity of our study. We have therefore chosen to keep the discussion focused on the experimentally supported PAR2-dependent pathway.

Reviewer #2 (Remarks to the Author):

The study by Meloun et al. identifies an as-yet-unknown sensor of allergens with protease activity and an important actor of type 2 immunity to allergens, namely atypical macrophages expressing PAR2 and Ly6G. The experimental design is robust and appropriate, the experiments are very well executed, the amount of data is impressive and they support most of the authors' conclusions. The manuscript is very well written and easy to read, and the authors

should be commended for their high-quality work. Since such Ly6G⁺ macrophages have not been previously described in the healthy mouse lung, I think that their characterization at steady-state warrants some additional investigation, especially regarding their localization. Comments from a to e below are related to this.

Major comments:

a. My understanding of the data in Fig. 3 is that Ly6G⁺ Macs at steady-state derive from GMPs postnatally and are not rapidly replaced by circulating monocytes. The authors use the word “tissue-resident” – what is their definition of tissue-residence? This should be clarified, as the possibility remains that they are intravascular (see below).

We appreciate the opportunity to clarify and expand on the biology and dynamics of Ly6G⁺ macrophages. In our manuscript, we do not use the term *tissue-resident* but instead refer to this population as *host-resident*, based on our parabiosis experiments demonstrating exclusive derivation from the host partner (**Fig. 3o**). At steady state, our data indicate that these macrophages arise postnatally from GMP progenitors (**Fig. 3q**) and are not rapidly replaced by circulating monocytes (**Fig. 3r**); however, they can be replenished by circulating monocytes following depletion (**Fig. 3s-t**).

They display high proliferative capacity, which is further enhanced upon allergen stimulation (**Extended Data Figs. 3c, 5g, 6d**). We detect these cells in the lung but not in circulation (**Extended Data Figs. 5h**). Intravascular labeling and immunofluorescence imaging show that they occupy a perivascular niche closely associated with pulmonary capillaries, rather than being intravascular (**Fig. 4i**). Collectively, these findings suggest that Ly6G⁺ macrophages are maintained locally within the lung and are not readily replenished by circulating monocytes under steady-state conditions, consistent with a self-renewing, long-lived population. As suggested by the reviewer, we have now included EdU pulse–chase experiments (see below) to further illustrate their proliferative dynamics and turnover within this stable population. In addition, we have summarized these findings in the discussion to emphasize that Ly6G⁺ macrophages likely represent a long-lived, self-renewing population in the lung that is poorly replenished from the circulation (see lines **545-548**).

b. Related to this, it seems that Ly6G⁺ macs are relatively long-lived. EdU-pulse experiments could provide insights into their half-life.

Thank you for this valuable suggestion. As recommended, to further investigate the long-lived maintenance and proliferative dynamics of Ly6G⁺ macrophages, we performed an EdU pulse-chase experiment. We found that Ly6G⁺ MΦ retained EdU labeling over time, although fluorescence intensity gradually declined, consistent with active proliferation within a stable population. In contrast, monocytes rapidly lost EdU signal, indicating high turnover and replacement by newly generated cells (**Extended Data Fig. 3d**; lines **232–237**). These findings support the conclusion that Ly6G⁺ macrophages represent a relatively long-lived, self-renewing population in the lung.

c. If they derived postnatally from GMPs, as shown with the Ms3a3Tdtom mice, when do they appear in the lung after birth?

As suggested, we examined the early postnatal appearance of Ly6G⁺ macrophages in the lung. We found that Ly6G⁺ MΦ were already detectable by postnatal day 7 (P7), indicating that they arise early after birth (**Extended Data Fig. 3i**; lines **253–254**).

d. In Fig. 3y, an important missing control is the chimeric data in non papain-injected mice. The authors state that accumulation of Ly6G⁺ macs in the lung after treatment with protease allergens requires PAR2, but they gate on papain⁺ cells, not all the cells, and they only look at treated mice, not naïve mice.

We agree that it is important to distinguish allergen-induced accumulation from potential baseline differences. As shown in our data, the presence of Ly6G⁺ macrophages in the lung is comparable between Par2^{-/-} and wild-type mice under steady-state conditions, indicating no difference in this population at baseline (**Extended Data Fig. 3j**). However, following papain administration, Par2^{-/-} mice fail to accumulate Ly6G⁺ macrophages in the lung (**Extended Data Fig. 3j**). Similarly, allergen-bearing Ly6G⁺ macrophages were markedly reduced in Par2^{-/-} mice after papain treatment compared with wild-type controls (**Fig. 3w**). Together, these data indicate that accumulation of Ly6G⁺ macrophages in response to protease allergens requires PAR2. In competitive 50% WT : 50% Par2^{-/-} bone marrow chimeras, papain⁺ Ly6G⁺ macrophages derived from WT donors selectively accumulated after papain challenge, whereas Par2^{-/-}-derived cells did not (**Fig. 3y**). These findings further confirm that PAR2 signaling is required in a cell-intrinsic manner for the allergen-induced accumulation of Ly6G⁺ macrophages in the lung.

e. My main conceptual concern relates to the localization of these cells. The authors state that Ly6G⁺ Macs are “perivascular” based on anti-CD45 iv staining, then later they say that Nr4a1+ Ly6G⁺ Macs are primarily present in the alveolar interstitium, located between the alveolar epithelium and the CD31+ capillary endothelium. First, to me, “perivascular” would rather refer to their preferential localization around bigger blood vessels (arterioles, venules), not capillaries, while the alveolar interstitium is not “perivascular” per se. I think that this is particularly important to clarify since it also relates to the site of sensitization to protease allergens and DC uptake? Does it occur in the alveoli? Or rather at the level of bronchial epithelium, which is suggested in many review and original articles in the field? Second, I understand that the confocal microscopy pictures shown in Extended Data Fig. 4c are taken from a naïve mouse lung. If correct, Nr4a1+Ly6G⁺ “Macs” are strikingly abundant on the slide, while they are a very rare population. An alternative, more likely, scenario is that the Nr4a1+Ly6G⁺ lung cells are lung neutrophils, which also express Nr4a1 (unpublished observations – and this does not fit with the data shown in Extended Data Fig. 4b – are Neu from the lung, as stated? or from the blood?). Third, it is impossible to draw the conclusion, based on the pictures shown, that Ly6G⁺ Macs are not intravascular, especially since they stain positive for anti-CD45 iv. Hence, it is imperative to re-assess the localization of Ly6G⁺ Macs at steady-state using additional markers: the best would be the use of Cx3cr1 GFP mice, or alternatively, the use of CD68, with a particular focus at peribronchial/perivascular locations.

Our flow cytometry data demonstrate that Nr4a1⁺Ly6G⁺ cells in the lung correspond precisely to the Ly6G⁺ macrophage population of interest (**Extended Data Fig. 6a–b**). Regarding the concern that these cells might represent neutrophils, our Nr4a1-GFP reporter analysis (**Extended Data Fig. 6a–b**) shows that lung neutrophils do not express Nr4a1, whereas Nr4a1 expression is restricted to non-classical monocytes and Ly6G⁺ macrophages in the lung. Furthermore, flow cytometric (**Extended Data Fig. 6a**) and newly incorporated imaging analyses (**Extended Data Fig. 6c**) confirm that Nr4a1⁺Ly6G⁺ cells express CX3CR1, further distinguishing them from neutrophils and supporting their macrophage identity. Thus, this population does not contain neutrophils.

In addition, we have now included scRNA-seq data from sorted Nr4a1⁺Ly6G⁺ macrophages, which clearly demonstrate their macrophage lineage and expression of core macrophage genes including Csf1r, Cx3cr1, Nr4a1, Bcl2, Pparg, Tgfbr3, Il10ra, Klf2, Klf4, Clec4a1, Cebpb, Itga4, Itgal, Spn, Cd9, Cd36, Lair1, Trem14, Fcgr4, and Ace, as well as Arg2, Dhrs9, Dusp1, Hdc, Alox5, Ptgs2, Il1r2, Il1rn, Lmnb1, and Ly6g, defining them as a unique Ly6G⁺ macrophage population (**Extended Data Fig. 5**; lines **314-335**).

Our imaging data show that Nr4a1⁺Ly6G⁺ macrophages are closely associated with the capillary network within the lung parenchyma, perivascular in the sense of pericapillary rather than around larger arterioles or venules. To prevent confusion, we have clarified this terminology throughout the manuscript and now describe these cells as residing at the alveolar-capillary interface. Confocal imaging (now shown in **Extended Data Fig. 6c**) was performed after allergen exposure (24 h after papain treatment), when Ly6G⁺ macrophages are more abundant and activated. Although these cells are rare at steady state, they can also be detected in naïve lungs at lower frequency and occupy the same anatomical niche.

Our data further show that Ly6G⁺ macrophages are not intravascular. They are absent from the circulation, and anti-CD45 i.v. labeling combined with flow cytometry and confocal imaging demonstrates that they have access to blood antigens yet reside outside the vascular lumen, in close apposition to capillaries, consistent with a pericapillary, interstitial localization (see amplified inset in **Extended Data Fig. 6c**).

This localization places Ly6G⁺ macrophages strategically between the alveolar epithelium and capillary endothelium, within a region directly accessible to inhaled allergens. In our model, allergens reach nearly all alveolar macrophages (**Extended Data Fig. 3f**), confirming that the distal airspaces are highly exposed to inhaled material and likely represent regions where allergens can readily reach the underlying tissue, given the thinness and permeability of the alveolar barrier. While protease allergens can also access the bronchial epithelium, our data specifically define the anatomical localization of Ly6G⁺ macrophages near the alveolar region, in close association with the capillary network. This positioning places them at the alveolar-capillary interface, a site also reached by inhaled allergens in our experimental model.

Our data support a key role for Ly6G⁺ macrophages in promoting Th2 sensitization. These cells occupy an interstitial niche where they are ideally positioned to take up soluble allergens that traverse the epithelial barrier, sense protease activity, and interact with dendritic cells (**Extended Data Fig. 6h–i**, lines **419–425**), thereby orchestrating the initiation of Th2 immune responses.

Collectively, these data establish that Ly6G⁺ macrophages represent a distinct perivascular/pericapillary macrophage population positioned at the alveolar–capillary interface. We have revised the manuscript to clarify this localization and included new supporting data (**Extended Data Fig. 6c**, inset showing Ly6G⁺ macrophages adjacent to CD31⁺ capillaries and co-localization of CX3CR1 with Nr4a1⁺Ly6G⁺ cells in the lung).

f. All the “data not shown” should be included in the manuscript, even though I appreciate already the substantial amount of data.

All data previously referred to as “not shown” have now been included in the revised manuscript. Specifically:

- **Extended Data Fig. 1l-m:** Accumulation of IL-4⁺ OT-II cells in mLNs following HDM and HDM^{HI} treatment.
- **Extended Data Fig. 1n-o:** Th2 response to HDM in Par2^{-/-} mice.
- **Extended Data Fig. 1r-s:** Effect of the PAR2 antagonist AZ3451 on the accumulation of IL-4⁺ OT-II cells in mLNs.
- **Extended Data Fig. 2a-d:** Effect of the PAR2 antagonist AZ3451 on mDC migration.
- **Extended Data Fig. 6e:** Th2 response to HDM in se_2^{-/-} mice.
- **Extended Data Fig. 6g:** Frequency of papain⁺ cells in the lungs of se_2^{-/-} mice 24 h after treatment.

The only exception is the data from global Nr4a1^{-/-} mice (**see line 385**), which were not included because they are equivalent to those from se_2^{-/-} mice. The se_2^{-/-} line represents a super-enhancer knockout that prevents high-level Nr4a1 expression, while Nr4a1^{-/-} mice are global knockouts exhibiting an indistinguishable phenotype in Ly6G⁺ macrophages. We believe that adding these data would not provide additional insight, as they are redundant and would only make the manuscript denser without improving clarity. However, if the reviewers or editors consider them necessary, we would be glad to include them.

Comments on figures:

- Figure 2a–f: The authors aim to evaluate the role of HDM proteases in promoting mDC migration. Why was heat-inactivated HDM used instead of HDM pretreated with E-64, a selective inhibitor of cysteine protease activity? This alternative would more precisely address the role of cysteine proteases.

In our study, we used an HDM allergen model together with two complementary approaches to inactivate cysteine protease activity—heat inactivation (HDM-HI) and selective cysteine protease inhibition (HDM-E-64)—alongside well-characterized cysteine protease (C1A) allergens (HDM-derived Der p 1 and papain, which share an identical catalytic mechanism). These approaches were employed to evaluate the role of cysteine protease activity in type 2 inflammation, Th2-cell accumulation, antigen-specific T-cell priming, dendritic-cell migration, and Ly6G⁺ macrophage activation. Although not every experiment was performed with all HDM preparations or protease allergens, parallel data using HDM-E-64 and the cysteine proteases Der p 1 and papain consistently demonstrated that cysteine protease activity is the critical enzymatic function driving Th2 immunity in our model. Importantly, our results further show that PAR2 serves as the sensor of this cysteine protease activity, mediating the downstream responses that promote allergen-induced accumulation and activation of Ly6G⁺ macrophages, dendritic cell migration, and subsequent Th2-cell priming. Accordingly, we used heat-inactivated HDM in **Figure 2a–f** as a general control to abolish overall protease activity, while complementary experiments using HDM-E-64, Der p 1, and papain (shown in other figures) confirm that cysteine protease activity—and its sensing through PAR2—specifically mediate the observed effects. To streamline the presentation, papain was used as a model cysteine protease to mechanistically dissect this pathway.

- Extended Data Figure 2b: The gating on histograms appears to be suboptimal. Given the high autofluorescence of macrophages, it is recommended to use Fluorescence Minus One (FMO) controls to accurately define positive populations. Similarly, for EdU staining, cells from non-injected mice should be subjected to the Click reaction to serve as a proper negative control.

We appreciate the reviewer's concern; however, we would like to clarify that all flow cytometry data were acquired and analyzed using rigorous gating strategies and appropriate controls. Our

laboratory has extensive experience with flow cytometric analysis of diverse immune cell populations, including lung macrophages, and all gating thresholds were established based on well-defined controls. Furthermore, the expression of the markers identified by flow cytometry in Ly6G⁺ macrophages has been independently validated by three separate experiments involving both scRNA-seq and bulk RNA-seq analyses (**Extended Data Figs. 4–5 and Fig. 4**), which confirm the expression of the same defining markers. Therefore, we respectfully disagree that the histograms appear “suboptimal,” as the data were generated rigorously following established cytometric standards and are consistent across multiple orthogonal platforms.

- Figure 4i: Gating on density plots is preferable, as it facilitates clearer discrimination between positive and negative populations. Furthermore, cells from PBS-injected control mice should be included as a negative control for CD45 intravenous staining.

We thank the reviewer for this suggestion. All gating for CD45 intravenous staining was defined using PBS-injected control mice, which served as negative controls in each experiment. In the figure, we presented representative histograms rather than density plots to facilitate direct comparison of fluorescence intensity across conditions within the same axis range. Because discrimination is based on a single fluorescence channel, the gating outcome is identical whether displayed as histograms or density plots.

- Extended Data Figure 3a-b: the moDC transcriptome fits perfectly with the signature of lung interstitial macrophages (IM), as they express all the macrophages and IM-related markers (Mafb, Fcgr1, C1q, Mrc1, etc), but do not express DC-specific markers. Even though the discrimination between moDCs and IMs remain contentious, I am curious why the authors chose to name these cells moDCs.

We thank the reviewer for this thoughtful comment. We agree that the distinction between monocyte-derived dendritic cells (moDCs) and interstitial macrophages remains an area of active discussion in the field. Our classification is based on both functional and developmental evidence. Following allergen exposure, our flow cytometry analysis detects a strong induction and allergen uptake by Cx3cr1⁺F4/80⁺Ly6C⁺CD11b⁺ monocyte-derived dendritic cells that markedly upregulate antigen-presentation molecules (**Fig. 3j**).

Consistent with this, our scRNA-seq analysis of allergen-bearing cells identifies clusters annotated as moDCs that upregulate genes involved in antigen processing and presentation while co-expressing myeloid markers shared with monocytes and macrophages. Pseudotime analysis further links these clusters developmentally to Ccr2⁺ classical monocytes, supporting their derivation from recruited monocytes. We recognize that different interpretations of this population are possible; however, our evidence indicates that these cells represent monocyte-derived antigen-presenting cells induced by allergen exposure. We therefore chose the term “moDCs” to reflect both their monocyte origin and their strong antigen-presenting phenotype. This distinction is not the central focus of the manuscript, and readers can interpret the data accordingly if they prefer alternative terminology.

- Extended Data Figure 3e: Instead of gating on CD11b⁺⁺ cells, staining for Ly6G would provide a more accurate identification of the target population.

In the revised version, we have now included staining for Ly6G, which more precisely defines the Ly6G⁺ macrophage population (**Extended Data Fig. 5h**). The updated figure confirms that Ly6G⁺ macrophages are present in the lung but absent from the blood, supporting the accuracy of our previous gating strategy.

• Figure 4b: The authors demonstrate that Nr4a1 is necessary for the accumulation of Ly6G⁺ macrophages and ncMo. However, it should be considered that ncMo may drive the differentiation of Ly6G⁺ macrophages upon allergen exposure. Therefore, the observed reduction in Ly6G⁺ macrophages in these mice may be secondary to the absence of ncMo, rather than reflecting a direct requirement of Nr4a1 in Ly6G⁺ macrophages themselves. This should be clarified.

We thank the reviewer for this insightful comment. To address this point, we performed additional analyses to define the relationship between Ly6G⁺ macrophages and non-classical monocytes (ncMo). Integrated scRNA-seq profiling of isolated Ly6G⁺ macrophages and ncMo revealed that these two populations cluster separately, indicating distinct transcriptional identities (**Extended Data Fig. 5a–c**; see lines **314–335**). Although they share a core set of genes (*Csf1r*, *Cx3cr1*, *Nr4a1*, *Bcl2*, *Pparg*, *Tgfbr3*, *Il10ra*, *Klf2*, *Klf4*, etc.), Ly6G⁺ macrophages uniquely expressed genes associated with effector and inflammatory programs, including *Arg2*, *Dhrs9*, *Dusp1*, *Hdc*, *Alox5*, *Ptgs2*, *Il1r2*, *Il1rn*, and *Ly6g* (**Extended Data Fig. 5d**; see lines **314–335**).

Pseudotime trajectory analysis linked ncMo to Ly6G⁺ macrophages through two distinct differentiation branches: one enriched for activation, adhesion, angiogenesis, and cytokine/chemokine pathways (Branch 1), and another for metabolic and stress-response pathways (Branch 2) (**Extended Data Fig. 5e,f**; see lines **314–335**). Differentiation along Branch 1 was accompanied by increased G2/M-phase representation, consistent with active proliferation (**Extended Data Fig. 5g**; see lines **314–335**). These data indicate that Ly6G⁺ macrophages represent a specialized macrophage lineage developmentally related to, but transcriptionally and functionally distinct from, ncMo.

Functionally, we have already linked the defective Ly6G⁺ macrophage phenotype to reduced Th2 immunity rather than to any secondary effect of ncMo loss. In mixed bone marrow chimeras (WT:*Par2*^{-/-}, WT:*se_2*^{-/-}, and *Par2*^{-/-}:*se_2*^{-/-} combinations; **Fig. 5i**), Ly6G⁺ macrophages—uniquely dependent on both *Par2* and *Nr4a1**se_2* expression—were selectively impaired in *Par2*^{-/-}:*se_2*^{-/-} chimeras after papain challenge (**Fig. 5i,j**), whereas ncMo and other lung populations remained unaffected. This selective defect was accompanied by reduced mDC migration to the mLNs (**Fig. 5k**), impaired OT-II expansion (**Fig. 5l**), and decreased Th2-cell and eosinophil accumulation in the lung (**Fig. 5m,n**).

Together, these findings demonstrate that although Ly6G⁺ macrophages may be developmentally related to ncMo, they are transcriptionally and functionally distinct and uniquely require both *Nr4a1* and *Par2* intrinsically for their activation and accumulation after allergen exposure. The observed phenotype therefore reflects a direct requirement for Ly6G⁺ macrophages in mediating Th2 immunity to protease allergens, rather than an effect mediated by ncMo.

• Figures 4c–d: The authors conduct mixed bone marrow chimeras (WT/*se_2*^{-/-}) to assess allergen uptake by Ly6G⁺ macrophages in the absence of Nr4a1, showing reduced papain capture by Ly6G⁺ macrophages from *se_2*^{-/-} origin. To strengthen these findings, it would be important to report the chimerism levels in Ly6G⁺ macrophage population. If Ly6G⁺ macrophages require Nr4a1 for differentiation, then cells of *se_2*^{-/-} origin should fail to differentiate into Ly6G⁺ macrophages, and this underrepresentation may account for the reduced frequency of papain⁺ Ly6G⁺ macrophages observed in Figure 4c.

We thank the reviewer for this thoughtful comment. The degree of chimerism within the Ly6G⁺ macrophage population in WT:se_2^{-/-} mixed bone marrow chimeras was comparable between WT and se_2^{-/-} donor cells, both in the total Ly6G⁺ macrophage pool and within the allergen⁺ fraction after papain exposure. Importantly, allergen⁺ Ly6G⁺ macrophages constituted the majority (~75–80%) of the total Ly6G⁺ macrophages following papain treatment (**see Extended Data Fig. 3f**). Thus, the reduced frequency of se_2^{-/-}-derived allergen⁺ Ly6G⁺ macrophages does not reflect impaired antigen uptake but rather reduced accumulation of se_2^{-/-} Ly6G⁺ macrophages. Consistent with this, we found that Ly6G⁺ macrophages require Nr4a1 (Nur77) for their expansion, as papain-treated WT mice showed a significant increase in Ki-67 expression within Ly6G⁺ macrophages, indicating proliferation upon allergen-driven activation (**Extended Data Fig. 6d**). These findings suggest that Nr4a1 controls the proliferative accumulation of Ly6G⁺ macrophages in response to protease allergens, rather than their antigen uptake capacity. This conclusion is explicitly discussed in the main text.

Further suggestions for improvement can be found below.

- Across all experiments, the figure panels are very small and should be enlarged with a focus on the most important findings (some gating strategies could be moved to supplement) for better readability.

We thank the reviewer for this helpful suggestion. We agree that figure readability is important; however, due to space constraints, we have retained the current layout. We believe that all panels are clearly legible and that keeping the key data within the main figures allows readers to fully evaluate the primary findings without needing to consult supplementary materials.

- All the data converge to the possibility that ncMo give rise to Ly6G⁺ Macs. This is particularly interesting since a previous study identified CD16.2⁺ ncMo in the mouse lung, also expressing Ace and Plac8, like Ly6G⁺ Macs. This possibility could be discussed.

We thank the reviewer for this thoughtful comment. We also considered this possibility and examined the relationship between Ly6G⁺ macrophages and the previously described CD16.2⁺ ncMo population expressing Ace and Plac8. To address this, we performed GSEA using the published CD16.2⁺ ncMo transcriptional signature; however, we did not detect any enrichment of this signature within our Ly6G⁺ macrophage population. In contrast, we observed strong enrichment with the canonical ncMo signature as shown in **Figure 4b** and **4e**. As the CD16.2⁺ ncMo signature did not overlap with our Ly6G⁺ macrophage transcriptome, we prefer to refrain at this stage from directly linking these two populations.

- Recent findings have shown that (likely other) recruited Ly6G⁺ macrophages can play a role in alveolar repair following influenza virus infection, and such Ly6G⁺ macs depend on both GM-CSF and type 2 cytokine signaling. Since cysteine proteases have previously been reported to induce GM-CSF production in non-classical monocytes (ncMo) (Kaur et al., Cell Reports, 2021), and since alternative activation via IL4/13 pops up in the transcriptomic data, it would be a nice mechanistic addition to investigate the extent to which Ly6G⁺ macrophages rely on these signaling pathways. Alternatively, this possibility could be discussed.

We thank the reviewer for this thoughtful comment. We agree that examining the contribution of GM-CSF, type 2 cytokine, or other signaling pathways to Ly6G⁺ macrophage activation is an important direction for future work. This represents a long-term interest of our group but falls beyond the scope of the present study. In our dataset, Ly6G⁺ macrophages express a broad

cytokine receptor profile, including *Csf1r*, *Csf2rb*, *Tgfbr1*, *Tgfbr2*, *Tgfbr3*, *Il10ra*, *Il2rg*, *Il4ra*, *Il13ra1*, *Il27ra*, and *Il31ra*, suggesting potential responsiveness to M-CSF, GM-CSF, TGF- β , IL-10, IL-4/IL-13, IL-27, and IL-31 signaling pathways. This expression pattern supports the idea that Ly6G⁺ macrophages are equipped to integrate multiple regulatory and remodeling cues within the lung microenvironment. We are currently further exploring the functional relevance of these cytokine–receptor pathways, which lies beyond the scope of the present study. We have noted this point in the Discussion (**lines 557–560**), highlighting that Ly6G⁺ macrophages display IL-4/IL-13–associated activation and remodeling signatures that warrant further investigation.

Reviewer #3 (Remarks to the Author):

Meloun and colleagues report, for the first time, that in an allergic model of pulmonary inflammation, Ly6G⁺ Macrophages produce CysLTs via PAR2, which facilitates dendritic cell migration to mediastinal lymph nodes (mLN). This study presents several novel findings, including the presence of Ly6g⁺ Macrophages, similar to those reported in an IAV model of lung inflammation (REF 30). The activation of Ly6G⁺ macrophages through Nr4a1 and PAR to produce CysLTs. And the effects of CysLTs on DCs migration to mLN to activate T-cells. The methodology includes the use of various chimeras and single-cell sequencing. The data presented support the main conclusions. I offered several observations and recommendations to further refine and expand the study.

The authors note in the discussion that lung Ly6g⁺ macrophages have previously been identified (Ref. 30). There are reported similarities between the Ly6G⁺ macrophages described in this study and those in Ref. 30 identified in a viral model of lung inflammation, including their origin from GMP precursors and their alveolar location. The authors may wish to consider investigating additional potential similarities, such as whether papain-induced Ly6g⁺ macrophages express Arg1. Are they short-lived, meaning are they only involved in sensitization? Additionally, how CysLTs enhance DC migration in response to chemokines?

We thank the reviewer for these insightful suggestions. We acknowledge in the Discussion that there may be similarities between the Ly6G⁺ macrophages described here and those reported in Ref. 30 (now Ref. 31 in the revised manuscript). However, when we explored potential overlaps, we did not observe enrichment of the transcriptomic signature of the previously described population in our dataset (GSEA, not shown), indicating that these cells are not transcriptionally similar. In our data, Arg1 is highly and uniquely expressed in the moDC population (cluster 3 in Extended Data Fig. 4a; cluster 0 in Fig. 4a). Although these data are not explicitly shown in the figures, they are available in the deposited GEO dataset. In contrast, Ly6G⁺ macrophages do not express Arg1 as a feature marker but instead show high and specific expression of Arg2 (**new data shown in Extended Data Fig. 5d**). This gene is also highly represented in our bulk RNA-seq dataset, which, although not explicitly shown in the figures, is included in the data deposited in GEO. Thus, our analyses do not reveal obvious transcriptional similarities between the Ly6G⁺ macrophages described here and those reported in Ref. 30 (now Ref. 31).

Our results, including parabiosis experiments (**Fig. 3o**) and newly added EdU pulse–chase analyses (**Extended Data Fig. 3d**), indicate that lung Ly6G⁺ macrophages are long-lived and possess strong self-renewing and proliferative capacity (**Extended Data Figs. 3c, 5g, 6d**). Nonetheless, our data show that this population can be depleted by irradiation or possibly by severe tissue injury, as may occur in viral infection models, after which it is repopulated by monocytes. This may account for model-dependent differences between our findings and those

reported in Ref. 30 (now Ref. 31). Overall, although we acknowledge that certain features partly resemble the previously described Ly6G⁺ macrophages, we also observe key differences—including long-term persistence, proliferative capacity, and a distinct transcriptional profile—that likely reflect either unique macrophage subsets or divergent activation states arising from specific experimental contexts. This will require further investigation, but our data indicate that the Ly6G⁺ macrophages identified here are long-lived and transcriptionally distinct from the short-lived populations described in viral infection models. Thus, while we recognize some shared features, the aim of this study was to define the identity and function of the Ly6G⁺ macrophages described here under steady-state and allergen-challenge conditions, rather than to establish equivalence with previously reported populations.

Regarding CysLT-mediated dendritic-cell migration, our data show that CysLTs selectively enhance CCR7-dependent migration toward CCL21, but not CCL19, as demonstrated in transwell assays (**Fig. 6p**).

Figure 3w: The authors state that “papain⁺ Ly6G⁺ MΦ did not accumulate in the lungs of treated Par2^{-/-} mice compared to WT”; however, there appears to be only a 50% reduction. Furthermore, given that moDCs were also reduced in PAR2^{-/-}, is PAR2 necessary for cMo or non-classical Mo differentiation into Ly6G⁺ macrophages, or for their proliferation?

We thank the reviewer for this important comment. We agree that it is critical to distinguish allergen-induced accumulation from potential baseline differences. As shown in our data, the presence of Ly6G⁺ macrophages in the lung is comparable between Par2^{-/-} and wild-type mice under steady-state conditions, indicating no difference at baseline (**Extended Data Fig. 3j**). However, following papain administration, Par2^{-/-} mice fail to accumulate Ly6G⁺ macrophages in the lung (**Extended Data Fig. 3j**). Likewise, allergen-bearing Ly6G⁺ macrophages are markedly reduced in Par2^{-/-} mice compared with wild-type controls (**Fig. 3w**).

Our competitive 50% WT : 50% Par2^{-/-} bone marrow chimeras demonstrate that only WT-derived Ly6G⁺ macrophages accumulate after papain challenge, whereas Par2^{-/-}-derived Ly6G⁺ macrophages do not (**Fig. 3y**). These findings indicate that PAR2 signaling is required in a cell-intrinsic manner for the allergen-induced accumulation of Ly6G⁺ macrophages in the lung. While moDCs are also reduced in Par2^{-/-} mice, the competitive chimera experiments show that this effect is not cell-intrinsic, as both WT and Par2^{-/-} progenitors give rise to moDCs at comparable frequencies.

Importantly, our parabiosis experiments combined with allergen challenge show that Ly6G⁺ macrophages are almost exclusively host-derived (**Fig.3o**), in contrast to cMo and ncMo populations, which equilibrate between donor and host partners. This indicates that Ly6G⁺ macrophages are largely maintained locally and are not continuously replenished from circulating monocytes during allergen exposure. Thus, the loss of these cells in Par2^{-/-} mice reflects a failure of resident Ly6G⁺ macrophages to expand or persist in response to allergen, rather than a defect in monocyte recruitment or differentiation. Collectively, these data demonstrate that PAR2 signaling is specifically required within Ly6G⁺ macrophages for their allergen-induced accumulation.

Extended Data 3b. Although Ly6G⁺ macrophages and non-classical monocytes share certain genes, non-classical monocytes were not identified in the transcriptional signature clusters.

We thank the reviewer for this insightful comment. We show that Ly6G⁺ macrophages and non-classical monocytes (ncMo) share transcriptional features, as both populations express a conserved core signature. However, ncMo were not detected in our allergen-induced scRNA-seq dataset because the analysis was performed on purified allergen⁺ cells, and ncMo do not take up allergen in the lung under these conditions.

To directly address the relationship between ncMo and Ly6G⁺ macrophages, we have now performed additional integrated scRNA-seq analyses of isolated Ly6G⁺ macrophages and ncMo. These analyses revealed that the two populations cluster separately, indicating distinct transcriptional identities (**Extended Data Fig. 5a–c**; see lines 314-335). While they share a core gene set (*Csf1r*, *Cx3cr1*, *Nr4a1*, *Bcl2*, *Pparg*, *Tgfb3*, *Il10ra*, *Klf2*, *Klf4*, *Clec4a1*, *Cebpb*, *Itga4*, *Itgal*, *Spn*, *Cd9*, *Cd36*, *Lair1*, *Trem14*, *Fcgr4*, and *Ace*), Ly6G⁺ macrophages uniquely express a group of effector and inflammatory genes—including *Arg2*, *Dhrs9*, *Dusp1*, *Hdc*, *Alox5*, *Ptgs2*, *Il1r2*, *Il1rn*, and *Ly6g*—that distinguish them from ncMo (**Extended Data Fig. 5d**; see lines 314-335). Pseudotime trajectory analysis further linked ncMo to Ly6G⁺ macrophages through two developmental branches: one enriched for activation, adhesion, angiogenesis, and cytokine/chemokine pathways (Branch 1), and another enriched for metabolic and stress-response programs (Branch 2) (**Extended Data Fig. 5e,f**; see lines 314-335). Differentiation along Branch 1 was associated with increased representation of cells in G₂/M phase, consistent with active proliferation (**Extended Data Fig. 5g**; see lines 314-335).

Moreover, Ly6G⁺ macrophages uniquely require both *Nr4a1* and *Par2* in a cell-intrinsic manner. In mixed bone marrow chimeras (WT:*Par2*^{-/-}, WT:*se_2*^{-/-}, and *Par2*^{-/-}:*se_2*^{-/-} combinations; **Fig 5i**), Ly6G⁺ macrophages—dependent on both *Par2* and *Nr4a1* *se_2* expression—were selectively impaired in *Par2*^{-/-}:*se_2*^{-/-} chimeras after papain challenge, whereas ncMo and other lung populations remained unaffected (**Fig. 5i,j**). This selective defect was accompanied by reduced mDC migration to the mLNs (**Fig. 5k**), impaired OT-II expansion (**Fig. 5l**), and decreased Th2-cell and eosinophil accumulation in the lung (**Fig. 5m,n**). Thus, the defective Ly6G⁺ macrophage phenotype leads to reduced Th2 immunity, whereas ncMo remain unaffected.

Together, these findings demonstrate that although Ly6G⁺ macrophages may be developmentally related to ncMo, they are transcriptionally and functionally distinct and uniquely require both *Nr4a1* and *Par2* intrinsically for their activation and accumulation after allergen exposure. The observed phenotype therefore reflects a direct requirement for Ly6G⁺ macrophages in mediating Th2 immunity to protease allergens.

Line 364. It would be informative to review the data on HDM, even if just as Extended Data.

We thank the reviewer for this suggestion. The requested data have now been incorporated as **Extended Data Fig. 6e**, showing that Th2 cells fail to accumulate in the lungs of *se_2*^{-/-} mice compared to WT controls following sensitization and challenge with HDM. This addition is now referenced in the main text (**line 396**).

In the BM chimera with WT and *Alox5*^{-/-} BM donors, papain sensitization yields minimal lung LTC₄ (~5 ng/ml). Yet, in figure 6n (WT/A-) under the same conditions, LTC₄ levels reach about 40 ng/ml. What accounts for this difference?

We thank the reviewer for this observation. The mean LTC₄ levels in the experiment with WT and Alox5^{-/-} BM donors were approximately 20 ng/ml for the WT group, whereas those in the WT/A⁻ chimeras shown in **Fig. 6n** averaged around 40 ng/ml. These experiments were performed independently, and the variation observed falls within the expected range of inter-experimental variability for LTC₄ quantification in lung tissue.

Figure 6k. Ly6g⁺ macrophage counts are comparable in s/A^{-/-} chimeras and controls. However, does ALOX5 deficiency alter their activation, proliferation, or recruitment?

We thank the reviewer for this question. As shown in **Fig. 6k**, Ly6G⁺ macrophage accumulation in the lungs of s/A^{-/-} chimeras is comparable to that of controls, indicating that ALOX5 deficiency does not affect their overall accumulation, recruitment, or expansion. Taken together, our findings suggest that ALOX5 is not required for Ly6G⁺ macrophage accumulation but rather for their effector functions, including LTC₄ production and the promotion of dendritic-cell migration.

Line 455. The statement that the three CysLT receptors “have redundant functions” is inaccurate. CysLT1, CysLT2, and CysLT3 each preferentially bind LTD₄, LTC₄, and LTE₄, respectively, resulting in distinct effects based on cell type and receptor expression.

We thank the reviewer for this accurate clarification. We have revised the sentence to reflect that CysLT1R, CysLT2R, and CysLT3R have distinct but complementary roles in mediating CysLT signaling, rather than redundant functions. The corrected text now appears in lines **496–497**.

Figure 7. Asthmatic patients are already sensitized to allergens. Does AZD9898 given during the challenge phase suppress T-cell responses, and are these results relevant to real allergens like house dust mites (HDM)?

We thank the reviewer for this excellent and important comment. We have now performed additional experiments to address this point. To determine whether LTC₄ S inhibition could also prevent the development of type 2 inflammation in previously sensitized mice, we administered AZD9898 at the beginning of the challenge phase rather than during sensitization (**Fig. 7j** and **7l**). Strikingly, this later treatment produced similar effects, including reduced mDC migration and re-expansion of donor OT-II cells in the mLNs (**Fig. 7j,k**), and suppression of Th2 responses (**Fig. 7l,m**) and eosinophilia (**Fig. 7n**) in the lung. These findings demonstrate that LTC₄ S inhibition remains effective when administered after sensitization, thereby modeling the therapeutic setting of allergen-experienced patients (see lines **508-515** and **line 524**).

Although these new experiments were performed using papain as a representative cysteine protease allergen, our study directly links this mechanism to HDM-induced inflammation. As detailed in the manuscript, we used multiple HDM preparations—including heat-inactivated (HDM-HI) and cysteine-protease-inhibited (HDM-E-64) HDM—together with the purified HDM protease Der p 1 and papain, which share an identical catalytic mechanism. Across these parallel systems, inhibition or inactivation of cysteine protease activity consistently abrogated Th2-cell priming, dendritic-cell migration, and Ly6G⁺ macrophage activation. Thus, papain serves as a mechanistic model allergen that recapitulates the same PAR2-dependent, cysteine-

protease-driven pathway underlying type 2 inflammation in HDM exposure. Together, these findings highlight that LTC₄S inhibition targets a conserved effector pathway relevant to both model and clinically significant allergens such as HDM.

Does AZD9898 affect Ly6G⁺ macrophage numbers or activation?

AZD9898 treatment did not affect the number/accumulation of Ly6G⁺ macrophages in the lung.

Line 513: The authors discuss Ruscitti et al. (ref 30) and indicate differences but do not highlight similarities in Ly6G⁺ macrophages, especially in relation to their location and development. Including this information would provide readers with a more comprehensive understanding.

We again thank the reviewer for this helpful comment regarding potential similarities with previously reported Ly6G⁺ macrophages. We agree that it is important to acknowledge potential overlap between the Ly6G⁺ macrophages described in our study and those reported by Ruscitti et al. (now Ref. 31). As discussed previously, we clarified in the Discussion that, based on currently available data, the main shared feature between these populations is the expression of Ly6G. While both populations are found near the alveoli, the Ly6G⁺ macrophages described here localize predominantly to perivascular and pericapillary regions and display a long-lived, self-renewing phenotype, whereas those reported by Ruscitti et al. appear transient arising following tissue injury. In addition, their developmental trajectories and broader transcriptional profiles are distinct. Additional studies will be required to determine whether these populations are developmentally or functionally related, or whether the observed differences reflect model-specific activation states.

Minor

Line 349: add "Fig. 5b"

We thank the reviewer for noting this. We have now added the reference to **Fig. 5b** at the indicated location (now in **line 380**) in the revised manuscript.

Line 374: add "Fig. 5h"

We thank the reviewer for noting this. We have now added the reference to **Fig. 5h** at the indicated location (now in **line 405-406**) in the revised manuscript.

REVIEWERS' COMMENTS

Reviewer #1 comments were assessed by in house editors and considered to be addressed.

We thank the reviewer and the in-house editors for confirming that all comments have been satisfactorily addressed.

Reviewer #2 (Remarks to the Author):

The authors have performed a tremendous amount of work during these revisions. In my opinion, the manuscript is now fully acceptable for publication in Nature Communications, and the authors should be commended for their impressive work.

We thank the reviewer for the positive assessment and greatly appreciate their supportive comments.

Reviewer #3 (Remarks to the Author):

I would like to express my gratitude to the Authors for their thorough response to my inquiries and for expanding their dataset. I am particularly pleased to note the identification of Ly6G⁺ macrophages as long-lived cells and their developmental relationship with non-classical monocytes (ncMo).

I agree with Review 2 and the authors' decision to clarify the description of "perivascular" macrophages, specifying their localization to the alveolar-pericapillary region. This description should be consistently incorporated throughout the manuscript, including the title, to enhance clarity and precision.

We thank the reviewer for this helpful suggestion. The manuscript already included the recommended clarifications specifying that these macrophages localize to the alveolar-pericapillary region. To align the title with this anatomical precision, we have updated the title accordingly. Because the micro-anatomical details are fully described in the text, this targeted modification provides the most accurate localization: pericapillary rather than the broader term perivascular, while keeping the title concise and clear.

Figure 6. The finding that Ly6G⁺ macrophages produce LTC₄, which subsequently induces CCR7⁺ dendritic cells (DC) to migrate toward CCL21, represents an intriguing mechanism. Although this may extend beyond the scope of the current study, it would be interesting to investigate which receptor is specifically involved in the novel Ly6G⁺ macrophage-induced activation of DC.

Thank you for this thoughtful suggestion. In our study, we show that both LTC₄ and LTD₄ promote DC migration to the lymph nodes *in vivo* and that both equally potentiate CCR7-driven DC migration toward CCL21 *in vitro*. Because these mediators signal through CysLTRs and can engage both CysLTR1 and CysLTR2, it is likely that either receptor, or both, contribute to this pathway. However, given their overlapping receptor usage and the fact that our data already demonstrate functional enhancement of DC migration by both LTC₄ and LTD₄, further

delineating which receptor, either one or both, is responsible would add only limited mechanistic depth within the scope of the current study. While interesting, we consider this question beyond the primary focus of our manuscript.

Regarding Figure 3v, I appreciate the authors' response to my comment. However, I maintain that the phrase "did not accumulate" might be an overstatement, given the comparable reduction observed in DC.

Thank you for this clarification. In our study, Ly6G⁺ macrophages increase approximately two-fold 24h after allergen exposure compared to basal levels in naïve lungs. This allergen-driven accumulation is completely absent when protease activity is removed from the allergen or when the host cannot detect protease activity. Therefore, in this specific context, we use "did not accumulate" to indicate that Ly6G⁺ macrophages fail to undergo their characteristic allergen-induced expansion under protease-deficient or protease-insensitive conditions.

Regarding Figure 6k, while it may extend beyond the primary scope of this study, considering the potential autocrine effects of CysLTs, I am curious whether the absence of Alox5 influences the production of other molecules by these macrophages upon activation.

Thank you for this insightful comment. Our focus in this study was specifically to test the functional consequences of Ly6G⁺ macrophage-derived cysteinyl leukotrienes on dendritic cell migration and subsequent Th2 priming. We demonstrate a direct effect of these macrophage-derived CysLTs on enhancing CCR7⁺ DC migration toward CCL21. While Alox5 deficiency could theoretically alter additional activation-associated mediators, determining broader autocrine or paracrine effects falls outside the scope of this work. Our data directly address the key mechanistic link we aimed to test: the requirement for Ly6G⁺ macrophage-derived CysLTs in promoting DC migration and Th2 initiation.